

# Open Source Algorithm for Detecting Sea Ice Surface Features in High Resolution Optical Imagery

Nicholas C. Wright[1], Christopher M. Polashenski[1,2]

[1]Thayer School of Engineering, Dartmouth College, Hanover, NH, USA

[2]U.S. Army Cold Regions Research and Engineering Laboratories, Hanover, NH, USA

*Correspondence to:* N. C. Wright (ncwright.th@dartmouth.edu)

**Abstract.** Snow, ice, and melt ponds cover the surface of the Arctic Ocean in fractions that change throughout the seasons. These surfaces control albedo and exert tremendous influence over the energy balance in the Arctic. Increasingly available m- to dm-scale resolution optical imagery captures the evolution of the ice and ocean surface state visually, but methods for quantifying coverage of key surface types from raw imagery are not yet well established. Here we present an open source system designed to provide a standardized, automated, and reproducible technique for processing optical imagery of sea ice. The method classifies surface coverage into three main categories: Snow and bare ice, melt ponds and submerged ice, and open water. The method is demonstrated on imagery from four sensor platforms and on imagery spanning from spring thaw to fall freeze-up. Tests show the classification accuracy of this method typically exceeds 96%. To facilitate scientific use, we evaluate the minimum observation area required for reporting a representative sample of surface coverage. We provide an open source distribution of this algorithm and associated training data sets and suggest the community consider this a step towards standardizing optical sea ice imagery processing. We hope to encourage future collaborative efforts to improve the code base and to analyze large datasets of optical sea ice imagery.

## 1 Introduction

The surface of the sea ice-ocean system exhibits many different forms. Snow, ice, ocean, and melt ponds cover the surface in fractions that change throughout the seasons. The relative fractions of these surfaces covering the Arctic ocean are undergoing substantial change due to rapid loss of sea ice (Stroeve et al., 2012), increase in the duration of melt (Markus et al., 2009; Stroeve et al., 2014), decrease in sea ice age (Maslanik et al., 2011), and decrease in sea ice thickness (Kwok and Rothrock, 2009; Laxon et al., 2013) over recent decades. As a whole, the changes are reducing albedo and enhancing the absorption of solar radiation, triggering an ice albedo feedback (Curry et al., 1995; Perovich et al., 2008; Pistone et al., 2014). Large-scale remote sensing has been instrumental in documenting the ongoing change in ice extent (Parkinson and Comiso, 2013), thickness (Kurtz et al., 2013; Kwok and Rothrock, 2009; Laxon et al., 2013), and surface melt state (Markus et al., 2009). An increasing focus on improving prediction of future sea ice and climate states, however, has also created substantial interest in better observing, characterizing, and modeling the *processes* that drive changes in albedo-relevant sea ice surface conditions such as melt pond formation, which occur at smaller length scales. For these, observations that resolve surface conditions explicitly are needed to understand the underlying causes of the seasonal and spatial evolution of albedo in a more sophisticated way.





Explicitly sensing the key aspects of the sea ice surface, including melt pond coverage, degree of deformation, floe
size, and lead distributions, requires evaluating the surface at meter to decimeter scale resolution. Variability in the
spatial coverage and morphology of these surface characteristics, however, occurs over hundreds of meters to tens of
kilometers. Estimates of aggregate scale surface coverage fraction must therefore be made at high resolution over
sample domains of many square kilometers. Quantifying the relative abundance of surface types over domains of
multi-kilometer scale from manned ground campaigns is both time consuming and impractical. Remote sensing
provides a more viable approach for studying these multi-kilometer areas. High resolution optical imagery (e.g. Figure
1) visually captures the surface features of interest, but the methods for analyzing this imagery remain under-
developed.
The need for remote sensing methods enabling quantification of meter-scale sea ice surface characteristics has
been well recognized, and efforts have been made to address it. Recent developments in remote sensing of sea ice
surface conditions fall into two categories: (1) methods using low-medium resolution satellite imagery (i.e. having
pixel sizes larger than the typical ice surface feature size) with spectral un-mixing type algorithms to derive aggregate
measures of sub-pixel phenomena (e.g. for melt ponds Markus et al., 2003; Rösel et al., 2012; Rösel and Kaleschke,
2011; Tschudi et al., 2008) and (2) methods using higher resolution satellite or airborne imagery (i.e. having pixel size
smaller than the typical scale of ice surface features) that is capable of explicitly resolving features (e.g. Inoue et al.,
2008; Kwok, 2014; Lu et al., 2010; Miao et al., 2015; Perovich et al., 2002; Renner et al., 2014; Webster et al., 2015).
The first category, those derived from low-medium resolution imagery, have notable strengths in their frequent
sampling and basin-wide coverage. They cannot, however, provide detailed statistics on the morphology of surface
conditions necessary for assessing our process-based understanding and have substantial uncertainty due to ambiguity
in spectral signal un-mixing. The second category – observations at high resolutions which explicitly resolve surface
properties – can provide these detailed statistics, but were historically limited by a dearth of data acquisitions. Recent
increases in imagery availability from formerly classified defense (Kwok, 2014) or commercial satellites (e.g.
DigitalGlobe), and increases in manned flights over the Arctic (e.g. IceBridge, SIZRS) have substantially reduced this
constraint for optical imagery. Likely increases in collection of imagery from UAV's (DeMott and Hill, 2016) and
increases in satellite imaging bandwidth (e.g. DigitalGlobe WorldView 4 launched in 2016) suggest that availability
of high resolution imagery will continue to increase.
Processing high resolution sea ice imagery to derive useful metrics quantifying surface state, however, remains a
major hurdle. Recent years have seen numerous publications demonstrating the success of various processing
techniques for optical imagery of sea ice on limited test cases (e.g. Inoue et al., 2008; Kwok, 2014; Lu et al., 2010;
Miao et al., 2015; Perovich et al., 2002b; Renner et al., 2014; Webster et al., 2015). None of these techniques, however,
have been adopted as a standard or been used to produce large-scale datasets, and validation has been limited.
Furthermore, none have been challenged by imagery collected across the seasonal evolution of the ice or used to
process data from multiple sensor platforms. These issues must be addressed to enable in large scale production-type
image processing and use of high resolution imagery as a sea ice monitoring tool.
A unique aspect of high resolution sea ice imagery datasets, which differs from most satellite remote sensing, is
the quantity of image sources and data owners. Distributed collection and data ownership means centralized processing



of imagery to produce a single product is unlikely. Instead, we believe that distributed processing by dataset owners
is more likely and the community therefore has a substantial need for a shared, standard processing protocol.
Successful creation of such a processing protocol would increase imagery analysis and result in the production of
datasets suitable for ingestion by models to validate surface process parameterizations. In this paper, we assess
previous publications detailing image processing methods for remote sensing and present a novel scheme that builds
from the strengths and lessons of prior efforts. Our resulting algorithm, the Open Source Sea-ice Processing (OSSP)
Algorithm, is presented as a step toward addressing the community need for a standardized methodology, and released
in an open source implementation for use and improvement by the community.

We began with three primary design goals that guided our development of the image processing scheme. The

method must (1) have a fully automatic workflow and have a low barrier to entry for new users, (2) produce accurate,
consistent results in a standardized output format, and (3) be able to produce equivalent geophysical parameters from
a range of disparate image acquisition methods. To meet these goals, we have packaged OSSP in a user-friendly
format, with clear documentation for start-up. We include a set of default parameters that should meet most user needs,
permitting processing of pre-defined image types with minimal set-up. The algorithm parameters are tunable to allow
more advanced users to tailor the method to their specific imagery input. We chose an open source format to enhance
the ability for the community to explore and improve the code relative to a commercial software. Herein, we discuss
how we arrived at the particular technique we use, and why it is superior to some other possible mechanisms. We then
demonstrate the ability of this algorithm to analyze imagery of disparate sources by showing results from high
resolution DigitalGlobe WorldView satellite imagery in both panchromatic and pansharpened formats, aerial sRGB
(standard Red, Green, Blue) imagery, and NASA Operation IceBridge DMS (Digital Mapping System) optical
imagery. In this paper, we classify imaged areas into three surface types: Snow and ice, melt ponds and submerged
ice, and open water. The algorithm is, however, suitable for classifying any number of categories, should a user be
interested in different surface types, and might be adapted for use on imagery of other surface types.
**2 Algorithm Design**
Two core decisions were faced in the design of this image classification scheme: (1) Whether to analyze the image by
individual pixels or to analyze objects constructed of similar, neighboring pixels, and (2) which algorithm to use for
the classification of these image units.

Prior work has shown that object-based classifications are more accurate than single pixel classifications when

analyzing high-resolution imagery (Blaschke, 2010; Blaschke et al., 2014; Duro et al., 2012; Yan et al., 2006). In this
case, 'high resolution' has a specific definition dependent on the relationship between the size of pixels and objects
of interest. An image is high resolution when surface features of interest are substantially larger than pixel resolution
and therefore are composed of many pixels. In such imagery, objects, or groups of pixels constructed to contain only
similar pixels (i.e. a single surface type), can be analyzed as a set. The m-dm resolution imagery meets this definition
for features like melt ponds and ice floes. Object based classification enables an algorithm to extract information about
image texture and spatial correlation within the pixel group; information that is not available in single pixel based



classifications and can enhance accuracy of surface type discrimination. Furthermore, object based classifications are
much better at preserving the size and shape of surface cover regions. Classification errors of individual pixel schemes
tend to produce a 'speckled' appearance in the image classification with incorrect pixels scattered across the image.
Errors in object based classifications, meanwhile, appear as entire objects that are mislabeled (Duro et al., 2012). Since
our intent is to process high-resolution imagery and produce measurements not only of the areal fractions of surface
type regions, but also to enable analysis of the size and shape of ice surface type regions (e.g. for floe size or melt
pond size determination), the choice of object based classification over pixel based was clear.

A wide range of algorithms were considered for classifying image objects. We first considered the use of

supervised versus an unsupervised classification schemes. Unsupervised schemes were rejected as they produce
inconsistent, non-intercomparable results. These schemes, examples of which include K-means clustering and
maximum likelihood classifiers, group observations into a predefined number of categories – even if not all feature
types of interest are present in an image. For example, an image containing only snow-covered ice will still be
categorized into the same number of classes as an image with snow, melt ponds, and open water together – resulting
in multiple classes of snow. Since the boundary between classes also changes in each image, standardizing results
across imagery with different sources and of scenes with different feature content would be challenging at best.

Supervised classification schemes instead utilize a set of known examples (called training data) to assign a

classification to unknown objects based on similarity to user-identified objects. Supervised classification schemes
have several advantages. They can produce fixed surface type definitions, allow for more control and fine tuning of
the algorithm, improve in skill as more points are added to the training data, and allow users to choose what surface
characteristics they wish to classify. While many machine learning techniques have shown high accuracy in remote
sensing applications (Duro et al., 2012), we selected a random forest machine learning classifier over other supervised
learning algorithms for its ability to handle nonlinear and categorical training inputs (Breiman, 2001; DeFries, 2000;
Pal, 2005), resistance to outliers in the training dataset (Breiman, 1996), and relative ease of implementation.

Our scheme, building on the success of Miao et al. (2015) in classifying aerial imagery, uses an image segmentation

algorithm to divide the image into objects which are then classified with random forest machine learning. We do not
attempt to assert that our method is the optimal method for processing sea ice imagery. Instead, we argue that it is
easily usable by the community at large, produces highly accurate and consistent results, and merits consideration as
a standardized methodology. In coordination with this publication, we release our code (available at
https://github.com/wrightni) with the intention of encouraging movement toward a standardized method. Our hope is
to continue development of the algorithm with contributions and suggestions from the sea ice community.
**3 Methods**
**3.1 Image Collection and Preprocessing**
The imagery used to test the algorithm was selected from four distinct sources in order to assess the algorithm's ability
to deliver consistent and intercomparable measures of geophysical parameters. We chose high resolution satellite
imagery from DigitalGlobe's WorldView constellation in panchromatic and 8 band multispectral formats, NASA



Operation IceBridge Digital Mapping System optical imagery, and aerial sRGB imagery collected using an aircraft-
mounted standard DLSR camera as part of the SIZONet project. We first demonstrate the technique's ability to handle
imagery representing all stages of the seasonal evolution of sea ice conditions on a series of 22 panchromatic satellite
images collected between March and August of 2014 at a single site in the Beaufort Sea: 72.0° N 128.0° W. We then
process 4 multispectral WorldView 2 images of the same site, each collected coincident with a panchromatic image
and compare results to assess the benefit of spectral information. Finally, we process a set of 20 sRGB images and 20
IceBridge DMS images containing a variety of sea ice surface types to illustrate the accuracy of the method on other
image sources.

The satellite images were collected by tasking WorldView 1 and WorldView 2 Digital Globe satellites over fixed
locations in the Arctic. Tasking requests were submitted to DigitalGlobe with the support and collaboration of the
Polar Geospatial Center. The panchromatic bands of WorldView 1 and 2 both have a spatial resolution of 0.46m at
nadir. The WorldView 1 satellite panchromatic band samples the visible spectrum between 400 nm and 900 nm, while
the WorldView 2 satellite panchromatic band samples between 450 nm and 850 nm. In addition, WorldView 2 has 8
multispectral bands at 1.84 m nadir resolution, capturing bands within the range of 400nm to 1040nm. Each
WorldView image captures an area of ~700-1300 km$^2$. Of the 22 useable panchromatic collections at the site, 15 were
completely cloud free while 7 of the images were partially cloudy. Images with partial cloud cover were manually
masked and cloud covered areas were excluded from analysis. The aerial sRGB imagery was captured along a 100km
long transect to the north of Barrow, Alaska with a Nikon D70 DSLR mounted at nadir to a light airplane during June
2009. The IceBridge imagery was collected in July of 2016 near 73° N 171° W with a Canon EOS 5D Mark II digital
camera. We utilize the L0 (raw) DMS IceBridge imagery, which has a 10cm spatial resolution when taken from 1500
feet altitude (Dominguez, 2010, updated 2017).

Each satellite image was orthorectified to mean sea level before further processing. Orthorectification corrects for
image distortions caused by off-nadir acquisition angles and produces a planimetrically correct image that can be
accurately measured for distance and area. Due to the relatively low surface roughness of both multiyear and first year
sea ice (Petty et al., 2016), errors induced by ignoring the real topography during orthorectification are small.
Multispectral imagery was pansharpened to the resolution of the panchromatic imagery. Pansharpening is a method
that creates a high resolution multispectral image by combining intensity values from a higher resolution panchromatic
image with color information from a lower resolution multispectral image. The pansharpened imagery used here was
created using a 'weighted' Brovey algorithm. This algorithm resamples the multispectral image to the resolution of
the panchromatic image, then each pixel's value is multiplied by the ratio of the corresponding panchromatic pixel
value to the sum of all multispectral pixel values. The orthorectification and pansharpening scripts were developed by
the Polar Geospatial Center at the University of Minnesota and utilize the GDAL (Geospatial Data Abstraction
Library) image processing tools (GDAL, 2016). All imagery used was rescaled to the full 8-bit color space for
improved contrast and viewing. No other preprocessing was done to the aerial sRGB imagery or IceBridge DMS
imagery.



### 3.2 Image Segmentation

A flow chart of the image processing steps taken after pre-processing is presented in Fig. 2. The first task in the image processing algorithm is to segment the image into groups of similar pixels, called objects. Accurate segmentation requires finding the boundaries between the natural surface types we wish to differentiate (e.g. the boundary between ice covered and open ocean), delineating their locations, and using these boundaries to produce image objects. Sea ice surface types have large differences in reflectivity and tend to change abruptly, rather than gradually over a large distance. We exploit this characteristic by using an edge detection algorithm to find boundaries between surface types. Figure 3 contains a visual demonstration of this process. First, a Sobel-Feldman operator (van der Walt et al., 2014) is applied to the input image (Fig. 3a). The Sobel-Feldman filter applies a discrete differentiation kernel across the image to find the local gradient of the image intensity. High gradient values correspond to abrupt changes in pixel intensity, which are likely boundaries between surface types. We scale the gradient values by an amplification factor of 2 in order to further highlight edge regions in the image. Following the amplification, we threshold the lowest 10% of the gradient image and set the values to zero. This reduces noise detected by the Sobel-Feldman filter, and eliminates weaker edges. The amplification factor and gradient threshold percentage are both tuning parameters, which can be adjusted to properly segment images based on the input image and the strength of edges sought.

The strongest edges in optical imagery of sea ice are typically the ocean-ice interface, followed by melt pond-ice boundaries, then ice ridges and uneven ice surfaces. In general, the more edges detected, the more segmented the image will become, and the more computational resources required to later classify the image objects. On the other hand, an under-segmented image may miss the natural boundaries between surfaces. Under segmentation introduces classification error because an object containing two surface types cannot be correctly classified. An optimally segmented image is one which captures all the natural surface boundaries with minimal over-segmentation (i.e. boundaries placed in the middle of features). The appropriate parameters for our imagery were tuned by visual inspection of the segmentation results. In such inspection, desired segmentation lines are manually drawn, and algorithm-determined segmentation lines are overlain and evaluated for completeness.

The result of the edge detection is a gradient map that marks the strength of edges in the image. We use a watershed segmentation technique to build complete objects based on edge locations and intensity (van der Walt et al., 2014). We first calculate all local minimum values in the gradient image, where a marker is then placed to indicate the origin of watershed regions. Each region then begins iteratively expanding in all directions of increasing image gradient until encountering a local maximum in the gradient image or encountering a separately growing region. This continues until every pixel in the image belongs to a unique set. With the proper parameter selection, each object will represent a single surface type. It is often the case that some areas will be over-segmented (i.e. a single surface feature represented by multiple objects). Over segmentation can either be ignored, or objects can be recombined if they meet similarity criteria in an effort to save computational resources. Here we chose to classify objects without recombination. Figure 3b shows the detected edges overlain on top of the input image.

The watershed segmentation algorithm benefits from the ability to create objects of variable size. Large objects are built in areas of low surface variability while many small objects are created in areas of high variability. This variable object sizing is well suited to sea ice surface classification because the variability of each surface type occurs





at different scales. Areas of open water and snow covered first year ice, for example, can often be found in large
expanses, while areas that contain melt ponds, ice ridges, or rubble fields frequently cover small areas and are tightly
intermingled with other surface types. Variable object sizes give the fine detail needed to capture surfaces of high
heterogeneity in their full detail, while limiting over segmentation of uniform areas.
**3.3 Segment Classification**
**3.3.1 Overview**
Once the image has been divided into regions of the same surface type, each object must be classified as to which
surface type it represents. We classify the objects using a random forest machine learning technique (Breiman, 2001;
Pedregosa et al., 2011). The development of a machine learning algorithm requires multiple iterative steps: 1) Select
attributes with which to classify each object, 2) create a training dataset, 3) classify unknown image objects based on
the training set, and 4) assess performance and refine, starting from step 1. Random forest classifiers excel for their
relative ease of use, flexibility in the choice of attributes that define each object, and overall high accuracy. The random
forest classifier is only one of many available machine learning approaches and others may also be suitable.
**3.3.2 Surface Type Definitions**
Another key challenge to quantitatively monitoring sea ice surface characteristics from high resolution imagery is a
lack of standardized surface type definitions. We noted above that high-resolution sea ice imagery comes from many
sources; each with different characteristics. As we will see below, each image source will need to have its own training
set created by expert human classifiers. The human classifier must train the algorithm according to definitions of each
surface type that are broadly agreed upon in the community for the algorithm to be successful in producing
intercomparable datasets. While at first the definitions of open water, ice and melt ponds might seem intuitive,
transitional states challenge these notions. Deciding where to delineate transitional states is important to
standardization. We have established the following definitions for the three surface types we sought to separate,
binning transitional states in a manner most consistent with their impact on albedo. (1) Open Water (OW): Applied to
surface areas that had zero ice cover as well as those covered by an unconsolidated frazil or grease ice. (2) Melt Ponds
(MP): Applied to surfaces where a liquid water layer completely submerges the ice. (3) Ice and Snow (I+S): Applied
to all surfaces covered by snow or bare ice, as well as decaying ice and snow that is saturated, but not submerged. The
definition of melt ponds includes the classical definition of melt ponds where meltwater is trapped in isolated patches
atop ice, as well as optically-similar ice submerged near the edge of a floe. We did not attempt to break these 'pond'
types because the distinction is unimportant from a shortwave energy balance (albedo) perspective. We further refined
the ice and snow category into two sub categories: (3a) Thick Ice and Snow, applied during the freezing season to ice
appearing to the expert classifier to be thicker than 50cm or having an optically thick snow cover and to ice during the
melt season covered by a drained surface scattering layer (Perovich, 2005) of decaying ice crystals and (3b) Dark and
Thin Ice, applied during the freezing season to surfaces of thin ice that are not snow covered including nilas and young
ice. This label was also applied during melting conditions to ice covered by saturated slush, but not completely
submerged in water. This is ice which in some prior publications (e.g. Polashenski et al., 2012) was labeled as 'slushy



bare ice'. We acknowledge that the boundary between the ice and snow sub-categories is often more a continuum than
a defined border but note that distinguishing the two types is useful for algorithm accuracy. Dividing the I+S type
creates two relatively homogeneous categories rather than a single larger category with large internal differences. A
user only interested in the categories of ice, ponds, and open water could simply re-combine them, as we have done
for analysis. Furthermore, we created a 'shadow' classification category that was used in panchromatic WorldView
images prior to melt onset. This classification category allowed the algorithm to differentiate dark shadows in spring
imagery from melt ponds in summer imagery – surface types that look similar based on single-band pixel intensity
values. The shadow category was grouped back with the I+S category for analysis.

### 3.3.3 Attribute Selection

Attributes are quantifiable measures of image object properties used by the classifier in discriminating surface types.
An enormous array of possible attributes could be calculated for each image object and could be calculated in many
ways. Examples of properties that could be quantified as attributes include values of the enclosed pixels, the size and
shape of the object, and values of adjacent pixels. The calculation of pixel values aggregated by image objects takes
advantage of the additional information held in the pixel group (as compared to individual pixels). We have compiled
a list representing a relevant subset of such attributes that can be used to distinguish different surface types in Table
1. We included a selection of attributes similar to those used in previous publications (e.g. Miao et al., 2015), as well
as attributes we have developed specifically for our algorithm.
Each image source provides unique information about the surface and it can be expected that a different list of
attributes will be optimal for classification of each image type – even though we seek the same geophysical parameters.
Calculating attributes of each image object is computationally expensive. We have, therefore, determined those that
are most valuable for classifying each image type to use in our classification. For example, pansharpened WorldView
2 imagery has 8 spectral bands which can inform the classification, while panchromatic versions of the same image
have only a single band. Our goal was to select a combination of attributes that describe the intensity and textural
characteristics of the object itself, and of the area surrounding the object. Table 1 indicates which attributes were
selected for use in classifying each image type.
We selected attributes by only including those with a high relative importance. The importance of each attribute
is a property of a random forest classifier, and is defined as the number of times a given attribute contributed to the
final prediction of an input. After initial tests with large numbers of attributes, we narrowed our selection by using
only those attributes that contributed to a classification in greater than 1% of cases. For discussion here, we group the
attributes into two broad categories: Those calculated using internal pixels alone (object attributes), and those
calculated from external pixel values (neighbor attributes).

### 3.3.4 Object Attributes

The most important attributes in the classification of an image segment were found to be aggregate measures of pixel
intensity within the object. We determine these by analyzing the mean pixel intensity of all bands and the median of
the panchromatic band. An important benefit of image segmentation is the ability to calculate estimates of surface





texture by looking at the variability within a group of pixels. The texture is often unique in the different surface types
we seek to distinguish. Open water is typically uniformly absorptive and has minimal intensity variance. Melt ponds,
in contrast, come in many realizations and exhibit a wider range in reflectance, even within individual ponds. To
estimate surface texture, we calculate the standard deviation of pixel intensity values and the image entropy within
each segment. Image entropy, H, is calculated as
$$H = - \sum p * \log_2 p$$

where $p$ represents the bin counts of a pixel intensity histogram within the segment. We also calculate the size of each
segment as the number of pixels it contains. We include image date as an attribute because sea ice surface
characteristics evolve appreciably, particularly before and after melt pond formation onset. Since date of melt onset
varies, the reader might argue that a more applicable attribute would be image melt state. Melt state, however, is not
an apriori characteristic of the image and would need to be manually defined, therefore not meeting our demand for a
fully automated scheme.

In multispectral imagery, we also calculate the ratios between the mean absorption of each segment in certain

portions of the spectrum. The important band ratios used for the multispectral WorldView imagery were determined
empirically. We tested every possible band combination, and successively removed the ratios that did not contribute
to more than 1% of segment classifications. In sRGB imagery we use the band ratios shown to be informative in this
application by Miao et al. (2015).

**3.3.5 Neighbor Attributes**

In addition to information contained within each segment, we utilize information from the surrounding area. To
analyze the surrounding region, we determine the dimensions of a minimum bounding box that contains the segment,
then expand the box by five pixels in each direction. All pixels contained within this box, minus those in the segment,
are considered to be neighboring pixels. Analogous to the internal attribute calculations, we find the average intensity
and standard deviation of these pixels. We also calculate the maximum single intensity within this region, which
measures for the presence of an illuminated neighboring ridge. The maximum neighboring intensity often provides
information to distinguish, for example, a shadowed ice surface from a melt pond. In panchromatic imagery, these
regions are often similar when evaluated solely on internal segment attributes. We do note that it is likely that a more
complex algorithm, for example identifying those pixels in a radius or distance to the edge of the segment, rather than
using a bounding box, would be more reliable. The tradeoff, however, is one of higher computational expense.

**3.4 Training Set Creation**

Four training datasets were created to analyze the images selected for this paper. One training set was created for

each imagery source: Panchromatic satellite imagery, multispectral satellite imagery, aerial sRGB imagery, and
IceBridge DMS imagery. Each training set consists of a list of image objects that have been manually classified by a
human and a list of attribute values calculated from those objects and their surroundings. The manual classification is
carried out by multiple sea ice experts. Experienced observers of sea ice can classify the majority (85%+) of segments
in a high resolution optical image with confidence. To address the ambiguity in correct identification of certain





segments, however, we used several (4) skilled sea ice observers to repeatedly classify image objects. For the initial
creation of our training datasets, two of the users had extensive training in the OSSP algorithm and surface type
definitions, while the other two had only a brief (i.e. <10 minute) introduction to the surface type definitions and no
experience with the algorithm. Figure 4 shows a confusion matrix to compare user classifications. Cells in the diagonal
indicate agreement between users, while off- diagonal cells indicate disagreement (Pedregosa et al., 2011). Agreement
between the two well-trained users was high (average 94% of segment identifications; Fig. 4a), while the agreement
between a well-trained user and a new user was lower (average of 86%; ig 4b). After an in-person review of the
training objects among all four users, the overall agreement rose to 97%. The remaining 3% of objects were cases
where the expert users could not agree on a single classification, even after review of the surface type definitions and
discussion. These objects were therefore not used in the final training set. Figure 5 shows a series of surface types that
span all our classification categories, including those where the classification is clear and those where it is difficult.
Difficult segments are over-represented in these images for illustrative purposes, and represent a relatively small
fraction of the total surface.
While the skill of the machine learning prediction increases substantially as the size of the training set grows,
creating large training sets is time consuming. We found that training datasets of approximately 1000 points yielded
accurate and consistent results. We have developed a graphical user interface (GUI) to facilitate the rapid creation of
large training sets (see Fig. 6). The GUI presents a user with the original image side by side with an overlay of a single
segment on that image. The user assigns a classification to the segment by visual determination.
The training dataset is a critical component of our algorithm because it directly controls the accuracy of the
machine learning algorithm – and using a consistent training set is necessary for producing intercomparable results.
In coordination with this publication we are releasing our version 1.0 training datasets with the intention that they
would represent a first version of *the* standard training set to use with each image type. Though we have found this
training dataset robust through our error analyses below, it is our intention to solicit broader input from the community
to refine and expand the training datasets available and release future improved versions.
In addition to cross-validating the creation of a training dataset between users, we assess the quality of our training
set through an out-of-bag (OOB) estimate, which is an internal measure of the training set's predictive power. The
random forest method creates an ensemble (forest) of classification trees from the input training set. Each classification
tree in this forest is built using a random bootstrap sample of the data in the training set. Because training samples are
selected at random, each tree is built with an incomplete set of the original data. For every sample in the original
training set, there then exists a subset of classifiers that do not contain that sample. The error rate of each classifier
when used to predict the samples that were left out is called the OOB estimate (Breiman, 2001). The OOB estimate
has been shown to be equivalent to predicting a separate set of features and comparing the output to a known
classification (Breiman, 1996).

**3.5 Assigning Classifications**

Once the training dataset is complete, the algorithm is prepared to predict the classification of unknown objects in the
images. The random forest classifier is run and a classified image is created by replacing the values within each





segment by the classification label predicted. Figure 3c shows the result of labeling image objects with their predicted
classification. From the classified image, it is possible to produce a number of useful statistics. The most basic
measurement is the total pixel counts for each of the three surface categories. This provides both the total area, in
square kilometers, that each surface covers, and the fraction of each image that is covered by each surface type. It
would also be possible to calculate measurements such as the average segment size for each surface, melt pond size
and connectivity, or floe size distributions. Each of these, however, has its own standardization problems significant
enough to merit their own paper.
For demonstration, we have used the output from our image classification to calculate the fractional melt pond
coverage for each date. The melt pond fraction was defined as the area of melt ponds divided by the total area covered
by ice floes, i.e.:
$$Melt\ Pond\ Coverage = \frac{Area_{MP}}{Area_{MP} + Area_{I+S}}$$

where the subscript MP indicates predicted melt ponds and I+S indicates predicted ice and snow.

### 3.6 Determining Classification Accuracy

The primary measure of classification accuracy was to test the processed imagery on a per pixel basis against human
classification. For each processed image, we selected a simple random sample of 100 pixels from the entire image and
asked four sea ice experts to assign a classification to those pixels. Note that in this case experts are asked to classify
individual pixels, rather than segments as they were asked to do in training set creation. For each image source, we
also selected one scene from which to check the classification of a larger sample of 1000 pixels. The larger sample
was created to demonstrate a tighter confidence interval in the accuracy, while the smaller samples were chosen to
demonstrate consistency across images. This metric gives a spatially weighted accuracy by assessing individual pixels
regardless of how the image was segmented. The pixels were presented to the user by showing the original image with
the given pixel highlighted. The observer then identified which of the three surface type categories best described that
pixel. This assignment is then compared to the algorithm's prediction without feedback to the human classifier. The
accuracy determined by each of the four observers was averaged to create a composite accuracy for each image.

### 4 Results

The OSSP image processing method proved highly suitable for the task of classifying sea ice imagery. A visual
comparison between the raw and processed imagery, shown in Fig. 7 can quickly demonstrate this in a qualitative
sense. Figure 7 contains two comparisons for each imagery source, selected to show the performance of the algorithm
on images that contain a variety of surface types. The colors shown correspond to the classification category; regions
colored black are open water, blue regions are melt ponds, gray regions are wet and thin ice, and white regions are
snow and ice. The quantitative processing results, including surface distributions and classification accuracy are shown
in Table 2. The overall classification accuracy was 96 ± 3% across 20 IceBridge DMS images; 95 ± 3% across 20





aerial sRGB images; 97 ± 2% across 22 panchromatic WorldView 1 and 2 images; and 98 ± 2% across 4 multispectral
WorldView 2 images.
The nature of the classification error is presented using a confusion matrix that compares the algorithm
classification with a manual classification of 1000 randomly selected pixels. One confusion matrix is shown in Fig. 8
for a single image from each of the four image sources. Values along the diagonal of the square are the classifications
where the algorithm and the human observer agreed, while values in off-diagonal areas indicate disagreement.
Concentration of error into a particular off-diagonal cell helps illustrate the types of confusion the algorithm
experiences. The number of pixels that fall into off-diagonal cells is low across all imagery types. In the IceBridge
imagery, there is a slight tendency for the algorithm to classify surfaces as open water where a human would choose
melt pond. This is caused by exceptionally dark melt ponds on the edge of melting through (Fig. 5, panels F and I).
Classification of mutlispectral WorldView imagery has a small bias towards classifying melt ponds over dark or thin
ice (Fig. 5, panel D). Aerial sRGB and Panchromatic WorldView images do not have a distinct pattern to their
classification errors.
The internal metric of classification training dataset strength, the Out of Bag Error (OOB) estimates, on a 0.0 to
1.0 scale, are shown in Table 3 for the trees built from our three training sets. The OOB estimate represents the mean
prediction error of the random forest classifier, i.e. an OOB score of 0.92 estimates that the decision tree would predict
92% of segments that are contained in the training dataset correctly.  The discrepancy between OOB error and the
overall classification accuracy is a result of more frequent misclassification of smaller objects; overall accuracy is area
weighted, while the OOB score is not.
**4.1 WorldView: Analyzing A Full Seasonal Progression**
We analyzed 22 images at a single site in the Beaufort Sea collected between March and August of 2014 to challenge
the method with images that span the seasonal evolution of ice surface conditions. The results of these image
classifications (shown in Fig. 9) illustrate the progression of the ice surface conditions in terms of our four categories
over the course of a single melt season.  While cloud cover impacted the temporal continuity of satellite images
collected at this site, we are still able to follow the seasonal evolution of surface features. A time series of fractional
melt pond coverage calculated from the satellite image site is plotted in Fig. 10. The melt pond coverage jumps to
22% in the earliest June image, as initial ponding begins and floods the surface of the level first year ice. This is
followed by a further increase to 45% coverage in the next few days. The melt pond coverage then drops back down
to 30% as melt water drains from the surface and forms well defined ponds. The evolution of melt pond coverage over
our satellite observation period is consistent with prior field observations (Eicken, 2002; Landy et al., 2014;
Polashenski et al., 2012) and matches the four stages of ice melt first described by Eicken (2002). The ice at this
observation site fully transitions to open water by mid-July, though it appears that the ice is advected out of the region
in the late stages of melt rather than completing melt at this location.



**5 Discussion**
**5.1 Error**
There are four primary sources of error in the OSSP method as presented, two internal to the method and two external.
Internal error is caused by segment misclassification and by incomplete segmentation (i.e. leaving pixels representing
two surface types within one segment). The net internal error was quantified in section 3.6 and 4. External error is
introduced by pixilation – or blurring of real surface boundaries due to insufficient image resolution – and human
error in assigning a 'ground truth' value to an aerial or satellite observation during training.
**5.1.1 Internal Error**
Through assessing the accuracy of each classified image on a pixel-by-pixel basis (section 3.6), we collect all internal
sources of error into one measurement: The algorithm either classified each pixel the same way as the human would
have, or it did not. Total internal accuracy calculated for the method, relative to human classifiers, is quite good, at
90-99% across all image types. Our experience is that this level of accuracy approaches the accuracy with which
fractional surface coverage can practically be determined from labor intensive ground campaign techniques such as
lidar and measured linear transects (e.g. Polashenski et al., 2012)
Misclassification error, the first type of internal error, occurs when the image classification algorithm fails to
replicate the human experts' decision-making process. This type of error is best quantified by analyzing the training
datasets. The OOB score for each forest of decision trees (Table 3) provides an estimate of each forest's ability to
correctly predict objects similar to those used to create the forest (section 3.4). The OOB score is not influenced by
segmentation error, because the objects selected for training dataset use were filtered to remove any objects that
contained more than one surface type. The most commonly misapplied category was the Dark and Thin Ice
subcategory of Ice and Snow. This category often represents surface types that are in a transitional state, and is often
difficult to classify even for a human observer.
Segmentation error, the second type of internal error, is caused when an object is created that contains more than
one of the surface types we are trying to distinguish. This occurs when boundaries between objects are not placed
where boundaries between surfaces exist; an issue most common where one surface type gradually transitions to
another. When this occurs, some portion of that object will necessarily be misclassified. We have compensated for
areas that lack sharp boundaries by biasing the image segmentation towards over-segmentation, but a small number
of objects still contain more than one surface type. During training set creation, we asked the human experts to identify
objects containing more than one surface type. 3.5% of objects were identified as insufficiently segmented in aerial
imagery, and 2% of objects in satellite imagery. This represents the upper limit for the total percentage of insufficiently
segmented objects for several reasons. First, segmentation error was most prevalent in transitional surface types (i.e.
Dark and Thin Ice), which represents a small portion of the overall image and is composed of relatively small objects.
This category is overrepresented in the training objects because objects were chosen to sample each surface type and
not weighted by area. In addition, insufficiently segmented objects are generally composed of only two surface types,





and end up identified as the surface which represents more of the object's area. Hence the total internal error introduced
by segmentation error is appreciably smaller than misclassification error, likely well under 1%.

### 455 5.1.2 External Error

The first form of external error is introduced by image resolution. At lower image resolutions, more pixels of the
image span edges, and smaller features are more likely to go undetected. Pixels on the edge of surface types necessarily
represent more than one surface type, but can be classified as only one. Misclassification of these has the potential to
become a systemic error if edge pixels were preferentially placed in a particular category. We assessed this error's
impact by taking high resolution IceBridge imagery (0.1m), downsampling to progressively lower resolution, and
reprocessing. Figure 11 shows the surface type percentages for three IceBridge images at decreasing resolution. Figure
12 shows a series of downsampled images and their classified counterparts. Surprisingly, despite clear pixilation and
aliasing in the imagery, little change in aggregate classification statistics occurred as resolution was lowered from 0.1
to 2m. This suggests that at resolutions used for this paper, edge pixels do not significantly impact the classification
results. It may also be possible to forego the pansharpening process discussed in section 3.1, and use 2m multispectral
WorldView imagery directly.

The second type of external error occurs when the human expert fails to correctly label a segment. Even skilled

human observers cannot classify every pixel in the imagery definitively, and indeed the division between the surface
types can sometimes be indistinct even to an observer on the ground. We addressed this concern by employing
observers extensively trained in the sea ice field, both in remote sensing and in-situ observations, comparing multiple
human classifications of the same segments. After discussion, the portion of image objects subject to human observer
disagreement or uncertainty is small. Human observers disagreed on 3% of objects creating our training sets. The
possibility of systemic bias among the expert observer classifications cannot be excluded because real ground truth,
in the form of geo-referenced ground observations from knowledgeable observers was, unfortunately, not available
for any of the imagery. Conducting this type of validation would be helpful, but given high confidence human expert
classifiers expressed in their classifications and low disagreement between them, may not be essential.

### 477 5.1.3 Overall Error

The fact that misclassification dominates the internal error metric suggests that error could be reduced if additional
object attributes used by human experts to differentiate surface types could be identified. The agreement between the
OSSP method and a human (96%+/-3%) is similar to the agreement between different human observers (97%),
meaning that the algorithm is nearly as accurate as a human manually classifying an entire image. If we exclude the
possibility for systemic error in human classification, and assume other errors are unrelated to one another, we can
calculate a total absolute accuracy in surface type determination as approximately 96%.

### 484 5.2 Producing Derived Metrics of Surface Coverage

The classified imagery, presented as a raster, (e.g. Fig. 7) is not likely to be the end product used in many analyses.
Metrics of the sea ice state in simpler form will be calculated. We already introduced the most basic summary metrics

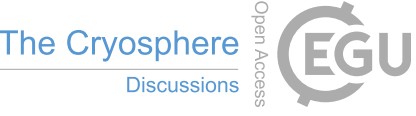

in section 4, where we presented fractional surface coverage calculated from the total pixel counts for each of the four
surface categories in each image. We also presented the calculation of melt pond coverage as a fraction of the ice-
covered portion of the image, rather than total image area. The calculation of these is straightforward. Other metrics
commonly discussed in the literature that could be produced include those capturing melt pond size, connectivity, or
fractal dimension; floe size distribution or perimeter to area ratio; and ridged ice coverage or frequency. As with
definitions of surface type, standardizing metrics will be necessary to produce intercomparable results. We discussed
the more complex metrics which could be derived from this imagery with several other groups. We determined that
standardizing these and other more advanced metrics will require more input and consensus building before a
community standard can be suggested. We leave determining standard methods for calculating these more complex
metrics to a future work.
For this general work, we felt that more important than the specific definition of additional metrics of surface
heterogeneity, is the consideration of what area must be imaged, classified, and summarized to constitute 'one
observation' and how representative such an observation is. Even with the increasing availability of high resolution
imagery, it is unlikely that high resolution imaging will regularly cover more than a small portion of the Arctic in the
near future. As a result, high resolution image analysis will likely remain a 'sampling' technique. Since the scale of
sea ice heterogeneity varies for each property type, a minimum area must be analyzed for a representative sample of
the surface conditions to be collected. Finding that minimum area involves addressing the 'aggregate scale' – the area
over which a measured surface characteristic becomes uniform and captures a representative average of the property
in the area (Perovich, 2005). Similarly, it may be possible to sub-sample within a representative area and determine
the mean of an aggregate scale sample within well constrained bounds, reducing processing time. Here we conduct
analysis of these sampling concepts and suggest analysis of this area be conducted for any metric.
Equipped with the images processed by OSSP, we sought to first determine the aggregate scale for the simple
fractional coverage metrics for ice and pond coverage (as a fraction of ice area). This would inform us, for example,
as to whether processing the entire area of a worldview image (~1000km$^2$) was necessary, or alternatively if a full
worldview image was sufficient to constitute a sample. We did this by evaluating the convergence of feature coverage
within image areas of increasing size to a regional mean. For each WorldView image acquired during the melt season,
we determined the fractional melt pond and ice coverage within non-overlapping gridded subsections. The size of
subsections was varied logarithmically from 100 x 100 pixels ($10^2$) to 31622x31622 pixels ($10^{4.5}$) or from 0.0025km$^2$
to 250km$^2$. For each subsample size, we gridded the image and evaluated every subsection within the entire image for
fractional surface coverage. Figure 10 shows a scatterplot of the fractional melt pond coverage determined from each
image subset plotted against the log of ice and pond area in the image subset. As the area sampled increases, the melt
pond fraction determined from independent sample areas within the overall image shows lower deviation from the
mean, as expected. To assist in evaluating the convergence toward the mean, we plot the 95% prediction interval for
each image subset size in Fig. 13a (large red dots). The range of pond fraction values between these two points
represents the interval within which 95% of samples of this size would fall. The size of the 95% prediction interval
declines linearly with respect to sample area in log space, with a slope of approximately 0.3 across most of the range
in sample area size explored. In other words, the prediction interval declines in width by 0.3 for each order of



magnitude the sample area is increased by. It appears that maximum convergence may have been reached or nearly
reached at a sample area of ~30km$^2$ (~$10^{1.5}$km$^2$), though we have an insufficient number of samples at this large area
size within a single image to be certain. Regardless of whether convergence is complete, the prediction interval tells
us that at this 30km$^2$ scale, 95% of image areas sampled could be expected to have pond coverage within 5% of the
mean of a full image (~1000km$^2$). This is consistent with prior work that indicated the aggregate scale for melt pond
fraction determination is on the order of several tens of square kilometers (Perovich, 2005; Perovich et al., 2002a),
and indicates that imagery representing an area as little as 3% of a Worldview image can provide an estimate of melt
pond fraction that is representative of the mean at 1000km$^2$ scale within what may be tolerable limits for many
applications. In Fig. 13b we conduct the same analysis, only this time for total ice-covered fraction (ponded +
unponded ice) of the image. We see the range of the prediction interval generally drops as larger samples are taken,
but does not converge as cleanly or quickly as the pond coverage prediction interval does - a finding that is unsurprising
since the ice fraction is composed of discrete floes with sizes much larger than melt ponds. (We limit prediction
interval to the range 0-1.) The limited convergence indicates that the aggregate scale for determination of ice covered
fraction is at least on the order of the scale of a WorldView image, and likely larger. Aggregate scale ice concentration,
unlike melt pond fraction, is a statistic better observed with medium resolution remote sensing platforms such as
MODIS or Landsat due to the need for a larger satellite footprint. WorldView imagery may be particularly useful for
determining smaller scale parts of floe size distributions or for validating larger scale remote sensing of ice fraction,
if the larger scale pixels can be completely contained within the worldview image. Floe size distribution will likely
require nesting of scales in order to fully access both large and small-scale parts of the floe size distribution.

We next investigated whether it is possible to further reduce the processing load required to determine the melt

pond or ice fraction of an image within certain error bounds by processing collections of random image subsets. In
this case, the idea is to collect a large number of random samples of from an image, instead of a single, larger sample
of the same area as the sum of the smaller random samples. We expected the random samples will better represent the
overall image mean because the single larger area is not composed of independent samples. Namely, ice conditions
are spatially correlated. We evaluated this hypothesis by processing sets of 100 image subsamples representing both
adjacent and randomly selected image areas. Results are shown in Fig 14. In Figure 14a, we plot a histogram of the
mean melt pond fraction determined from 1000 sets of image areas. Each of the sets contained 100 sample areas of
100x100 pixels. The means determined from sets that contained adjacent image areas, essentially representing a single
image sample 10 times larger in area, are in blue. The means determined from sets that contained randomly selected
image areas, are in red. Though both sets represent samples of the same total image area, the one composed of
independent subsets randomly selected from across the image does a much better job of representing the mean value.
Figure 14b shows the standard deviation for the same image sets. Independent samples from across the image show a
lower range in lower standard deviation within the image sets as well, though the average standard deviation is slightly
higher. Again, this is expected, given the strong spatial correlation of surface coverage fraction within the images.

We next test the central limit theorem to see how well we can predict the error bounds from processing a single

set of independent (i.e. randomly distributed) samples. The central limit theorem states that when taking the mean of
a sufficiently large number of independent samples of a random variable, the standard error of the mean of the samples



is equal to $\frac{\sigma}{\sqrt{N}}$ where σ is the standard deviation of the sample values and $N$ is the sample size. The standard deviation
of pond coverage fraction in sets of 100 sub-images ranged from 0.15 to 0.25 across the 1000 sample sets run (see
histogram in Fig. 14b) This yields a predicted standard error of the mean determined from any one of these sets of
0.015 to 0.025. The observed standard deviation in the mean values across all 1000 sample sets presented in Fig. 14a
is 0.0201, indicating that the central limit theorem applies in this case.
Returning to Fig. 13, we now place another set of 95% prediction interval bounds, this time representing twice the
standard error determined from the central limit theorem. These bounds represent the prediction interval for 100
randomly distributed sub-areas that total the area on the x axis. The result is quite powerful. We show that processing
a relatively small fraction of image area, so long as that sub-area is collected from a large number of samples randomly
distributed across the area, permits expedient determination of melt pond fraction within that image area with small
error bounds. If the total image is large enough, the value will be representative of the aggregate scale. In this case,
processing as little as 5km$^2$ (~0.5%) of the image permits determination of a mean that lies within 0.025 of the true
image mean 95% of the time. Also indicated on the plot is a 5% uncertainty band around the mean melt pond fraction
determined for the entire image. We estimate that 5% of the determined melt pond fraction is a reasonable estimate of
the sum of internal (2-4%) and external errors in our processing algorithm. For large scale processing, we suggest that
when the 95% prediction interval (sampling error) is well below the image processing technique accuracy, sampling
of larger areas is no longer worthwhile.
A similar analysis is presented in Fig. 14c and 14d for ice fraction. While the WorldView image is likely not large
enough to represent the aggregate scale for ice fraction, randomly sampling the image still provides an expedient way
to determine the mean ice fraction of the image within certain bounds, while processing only a small fraction of the
image. A test of the central value theorem again shows that it also applies in this case and provides a good estimate of
the error of a mean ice fraction calculated from a set of random sub images. The green dots again indicate the 95%
prediction interval that can be expected for image sets containing 100 samples that total the area on the x axis.
These explorations of image sampling permit us to recommend, with some safety factor built in, that users must
process imagery representing at least 5km$^2$ in surface area, selected in at least 100 randomly located subsets from
domains of at least 30 km$^2$ to produce an 'aggregate scale' estimate of pond coverage. We suggest a standard, which
incorporates some 'safety factor', for processing imagery to produce estimates of melt pond fraction should be to
process 10km$^2$ of area contained in at least 100 randomly located image subsets from domains of at least 100km$^2$. We
note that flying a UAV over a domain and collecting imagery along flight tracks will not count as fully 'random' in
this context, since the images along-track are spatially correlated. Since an image does not represent the aggregate
scale for ice fraction, we cannot recommend a specific sampling strategy for the aggregate scale, but note that
processing of 5km$^2$ of imagery from 100 subsets produces a prediction interval around the mean of approximately the
same size as the upper limit of uncertainty for our image processing technique. These recommendations should be
considered provisional, because they are subject to impacts from differences in ice property correlation scales, and
should be further evaluated for accuracy as larger processed datasets are available.



### 5.3 Community Adoption


We have provided a free distribution of the OSSP algorithm and the training sets discussed in section 3.4 and 4 as a
companion to this publication, complete with detailed startup guides and documentation. This OSSP algorithm has
been implemented entirely in Python using open source resources with release to additional users in mind. The code,
along with documentation, instructional guidelines, and premade training sets (those used for the analyses herein) is
available at https://github.com/wrightni. The software is packaged with default parameters and version controlled
training sets for 4 different imagery sources. The package includes a graphical user interface to allow users to build
custom training datasets that suit their individual needs. The algorithm was constructed with the flexibility to allow
for the classification of any number of features given an appropriate training dataset.
Our intention is that by providing easy access to the code in an open source format, we will enable both specific
inquiries and larger scale image processing that supports community efforts at general sea ice monitoring. We plan to
continue improving and updating the code as it gains users and we receive community feedback. We hope to encourage
others to design their own features and add-ons. Since the predictive ability of the machine learning algorithm
improves as more training data is added, we wish to strongly encourage the use of the GUI to produce additional
training sets and we plan to collate other users training sets into improved training versions. See documentation of the
training set creation GUI for more information on how to share a training set.
The OSSP algorithm helps to bring the goal of having a standardized method for deriving geophysical parameters
from high resolution optical sea ice imagery closer to reality. In the larger picture, developing such a tool is only the
first step. We recall that the motivation behind this development was the need to quantify sea ice surface conditions
in a way that could enable better understanding of the processes driving changes in sea ice cover. The value of the
toolkit will only be realized if it is used for these scientific inquiries. We look forward to working with imagery owners
to facilitate processing of additional datasets.

### 6. Conclusions


We have implemented a method for classifying the sea ice surface conditions from high resolution optical imagery of
sea ice. We designed the system to have a low barrier to entry, by coding it in an open source format, providing
detailed documentation, and releasing it publicly for community use. The code identifies the dominant surface types
found in sea ice imagery; open water, melt ponds, and ice, with accuracy that averages 96 percent – comparable to the
consistency between manual expert human classifications of the imagery. The algorithm is shown to be capable of
classifying imagery from a range of image sensing platforms including panchromatic and pansharpened WorldView
satellite imagery, aerial sRGB imagery, and optical DMS imagery from NASA IceBridge missions. Furthermore, the
software can process imagery collected across the seasonal evolution of the sea ice from early spring through complete
ice melt, demonstrating it is robust even as the characteristics of the ice features seasonally evolve. We conclude,
based on our error analysis, that this automatic image processing method can be used with confidence in analyzing
the melt pond evolution at remote sites.



With appropriate processing, high resolution imagery collections should be a powerful tool for standardized and
routine observation of sea ice surface characteristics. We hope that providing easy access to the methods and algorithm
developed herein, we will facilitate the sea ice community convergence on a standardized method for processing high
resolution optical imagery either by adoption of this method, or by suggestion of an alternate method complete with
code release and error analysis.

The authors declare that they have no conflict of interest.

*Data Availability.* The OSSP algorithm code is available from https://github.com/wrightni during the review process,
and will be transferred to a permanent repository for publication. Image data and processing results are available at
the NSF Arctic Data Center (ADC), and a permanent DOI is pending. Raw and preprocessed image data from
DigitalGlobe WorldView images will not be made available for copyright reasons, but can be acquired from
DigitalGlobe or the Polar Geospatial Center at the University of Minnesota.

*Acknowledgements.* This work was supported by the Office of Naval Research Award N0001413MP20144 and the
National Science Foundation Award PLR-1417436. We would like to thank Donald Perovich and Alexandra Arntsen
for their assistance in creating machine learning training datasets. We would also like to thank Arnold Song, Justin
Chen, and Elias Deeb for their assistance and guidance with the development of the OSSP code. WorldView satellite
imagery was provided with the DigitalGlobe NextView License through the University of Minnesota Polar Geospatial
Center. A collection of the aerial imagery was collected by the SIZONet project. Some data used in this paper were
acquired by NASA's Operation IceBridge Project.

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

**Figures**

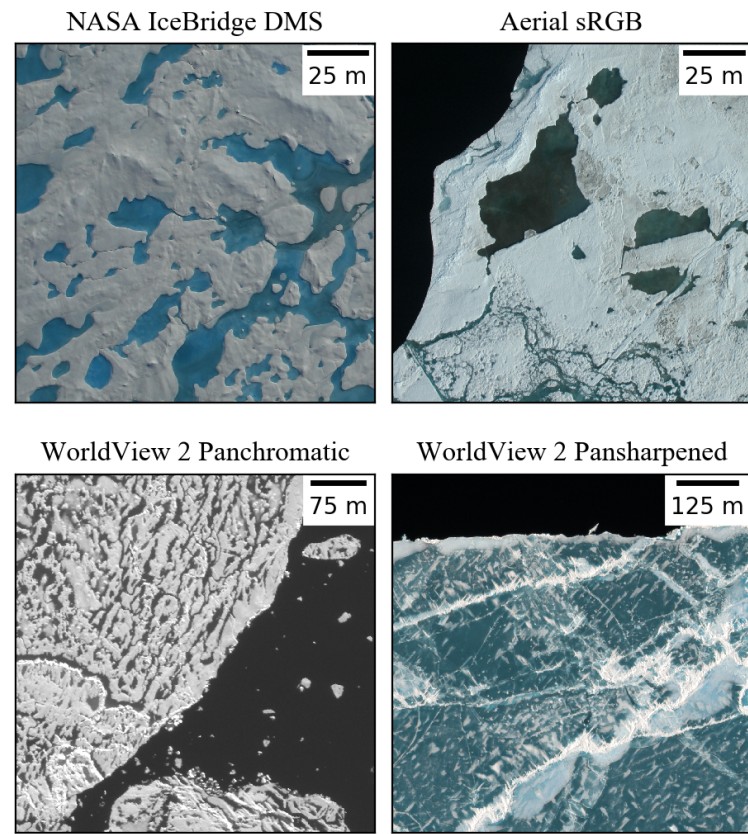


**Figure 1. Examples of imagery types we seek to process in this study. Note the varying imagery sources, resolutions, and**
**spectral information available for each image type.**





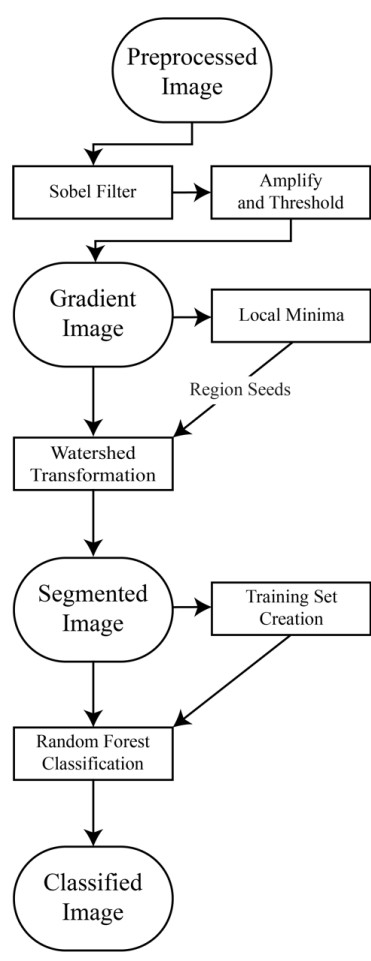


**Figure 2. Flow diagram depicting the steps taken to classify an image in the OSSP algorithm.**





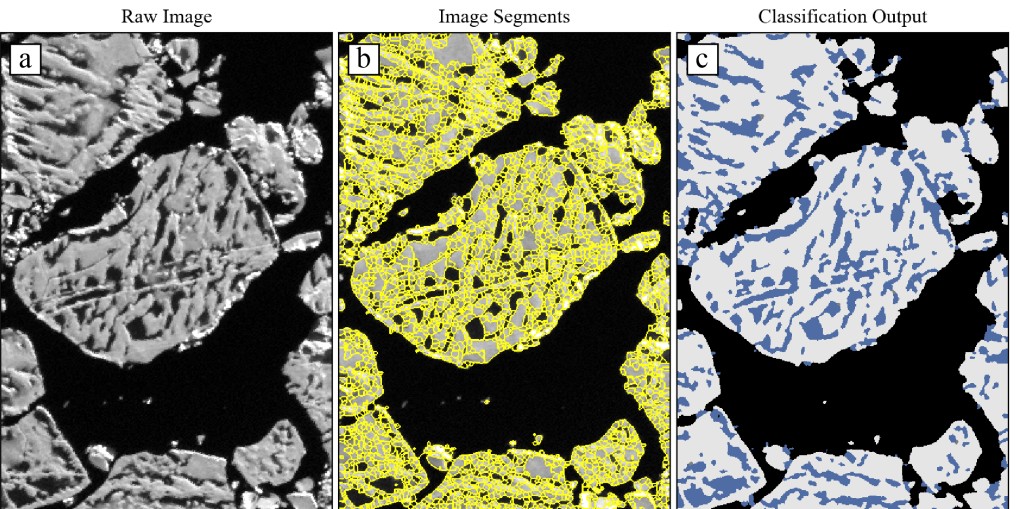

**Figure 3. Important steps in the image processing workflow. Panel (a) shows a section of a preprocessed panchromatic WorldView 2 satellite image, taken on July 1, 2014. Panel (b) shows the outline of image objects created from our edge detection and watershed transformation. Panel (c) shows the classified result after running each object through a random forest classifier.**

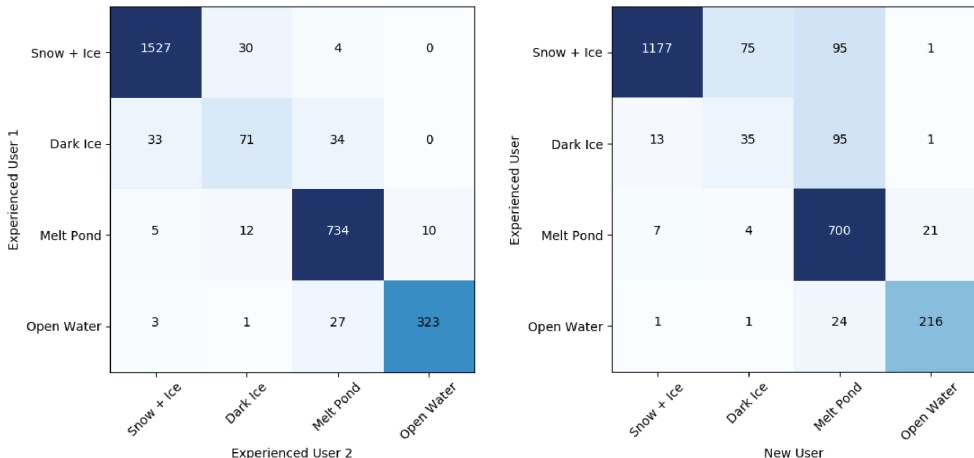

**Figure 4. Confusion matrices comparing classification patterns between two users experienced with the image processing algorithm (left) and between an experienced user and a new user (right). Squares are colored based on the number of pixels in that cell, with darker colors indicating a larger number of pixels.**







Figure 5. Examples of surfaces seen in aerial imagery of sea ice that span our four classification categories. Panel A: snow
covered surface. Panel B: Ice with a thin surface scattering layer where disagreement on true classification exists -
represents a small fraction of total surface area. Panel C: Panel D: Surface transitioning to a melt pond that is not yet fully
submerged. Panel E: Melt pond. Panel F: Dark melt pond that has not completely melted through. Panel G: Submerged
ice. Panel H: Brash, mostly submerged, included in the melt pond category. Panel I: Melt pond that has completely melted
through to open water. Panel J: Open water.

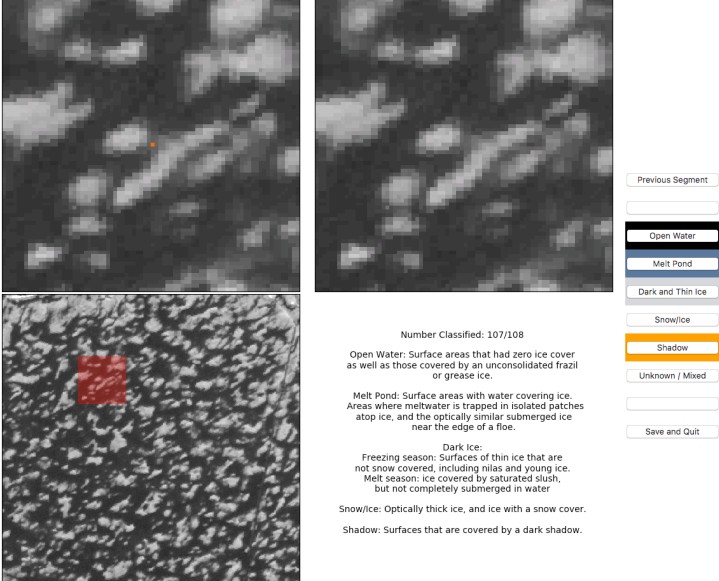


Figure 6. Graphical user interface for creating training datasets and assessing the accuracy of a classified image. As shown,
the user interface is demonstrating the classification of a single pixel for use in the overall accuracy assessments (section
773   3.6).





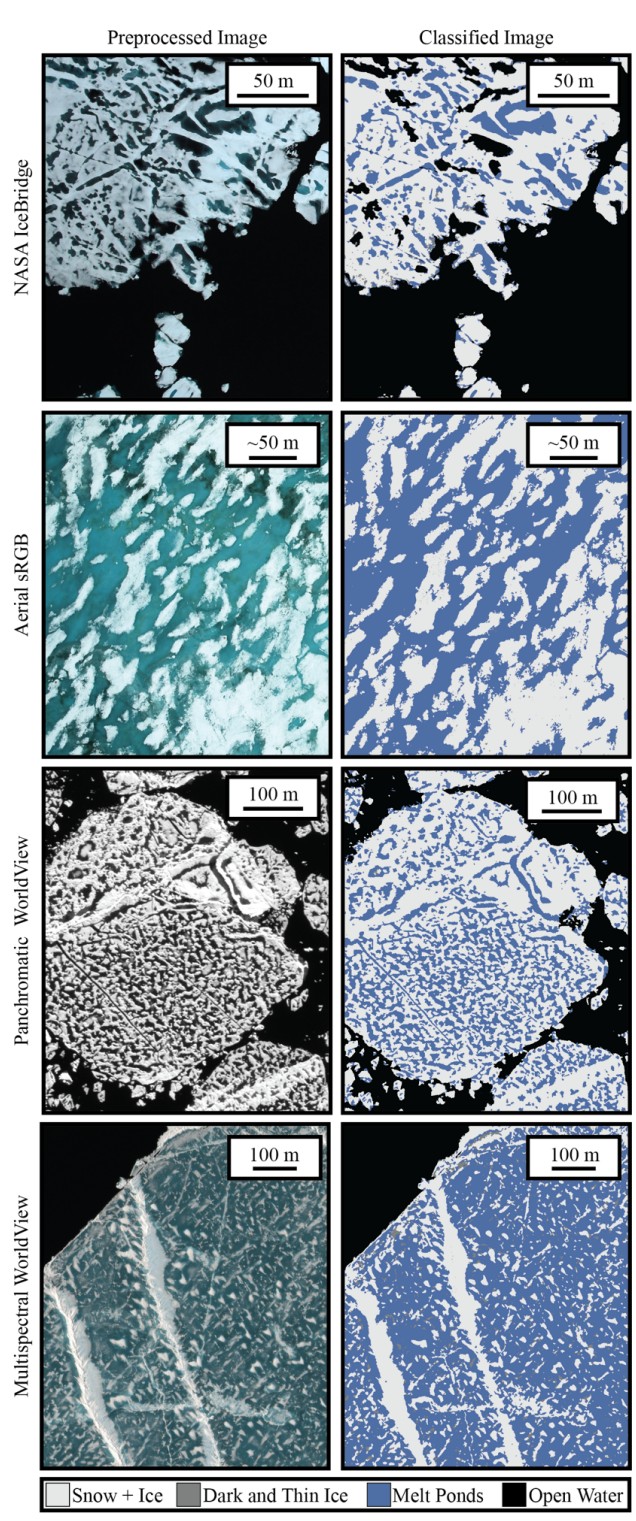



**Figure 7. Side-by-side comparison of preprocessed imagery and the classified result. One scene was selected from each imagery source. NASA IceBridge imagery is in very late stages of melt with many ponds having already melted through to the ocean.**

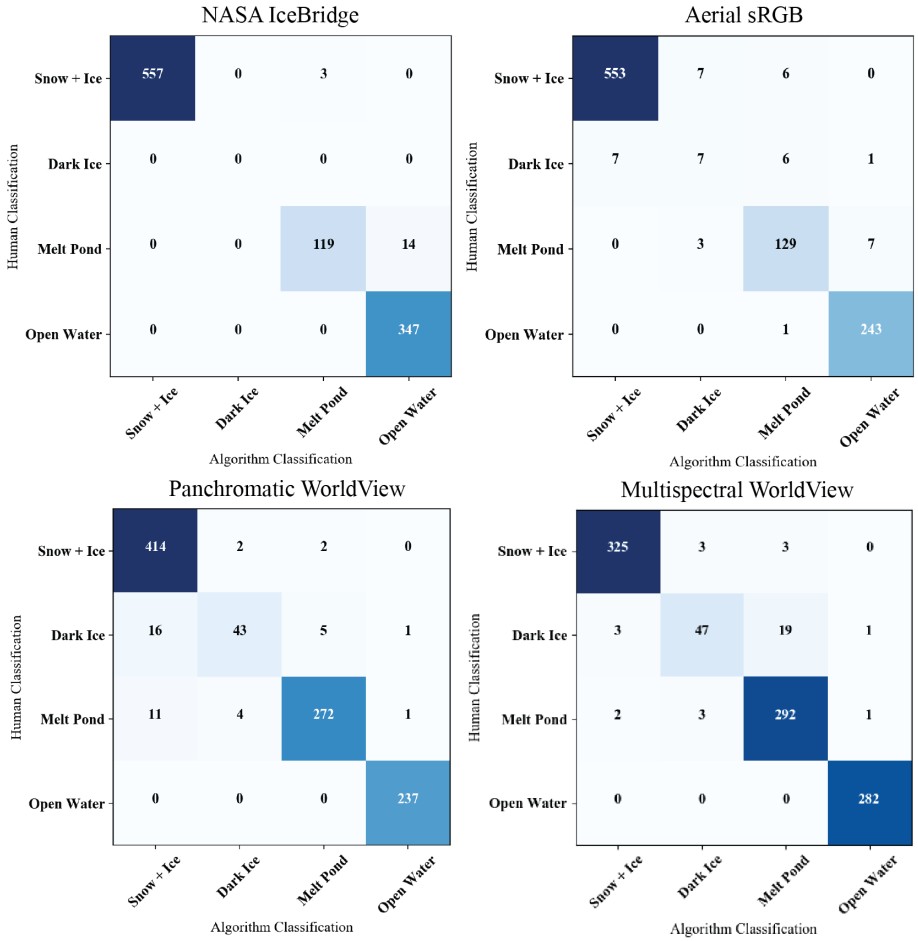

**Figure 8. 1000-pixel accuracy confusion matrix for each image type. Squares are colored based on the number of pixels in that cell, with darker colors indicating a larger number of pixels.**

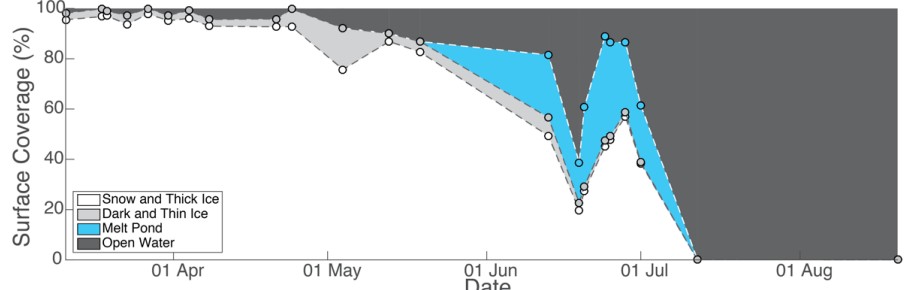



**Figure 9. Seasonal progression of surface type distributions at our satellite image collection site; 2014 in the Beaufort Sea**
**at 72°N 128°W.**

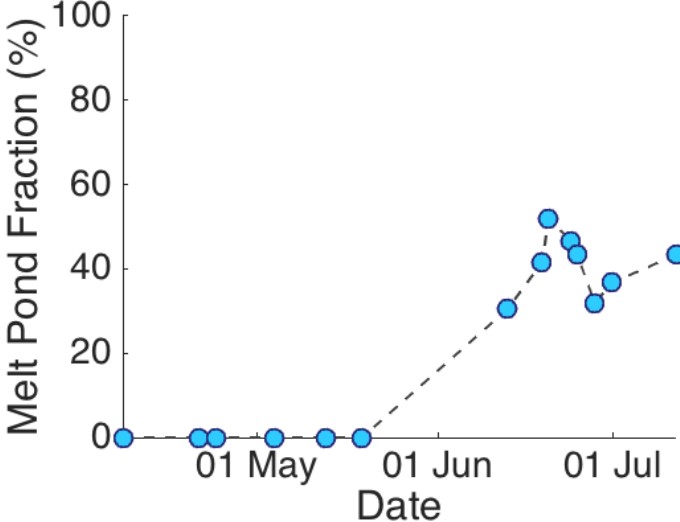


**Figure 10. Evolution of melt pond fraction over the 2014 season at our satellite image collection site; 2014 in the Beaufort**
**Sea at 72°N 128°W.**




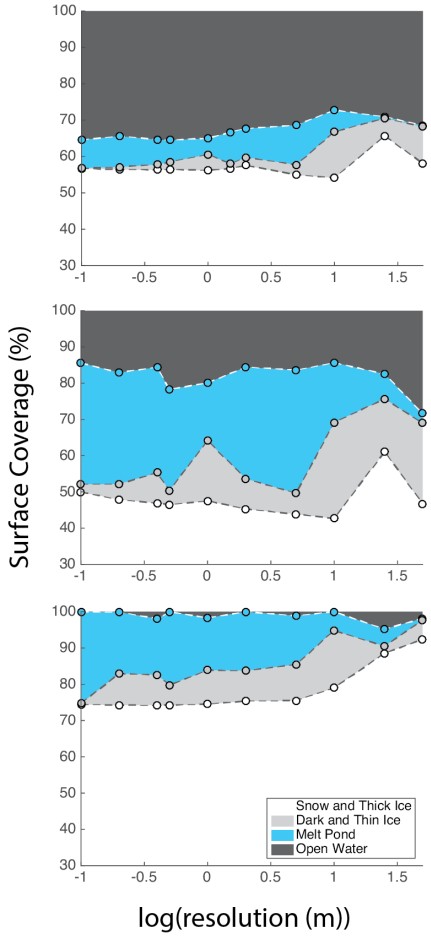


**Figure 11. Change in surface coverage percentage as a result of downsampling IceBridge imagery. Imagery starts at the nominal IceBridge resolution of 0.1m and is degraded to a maximum of 50m.**





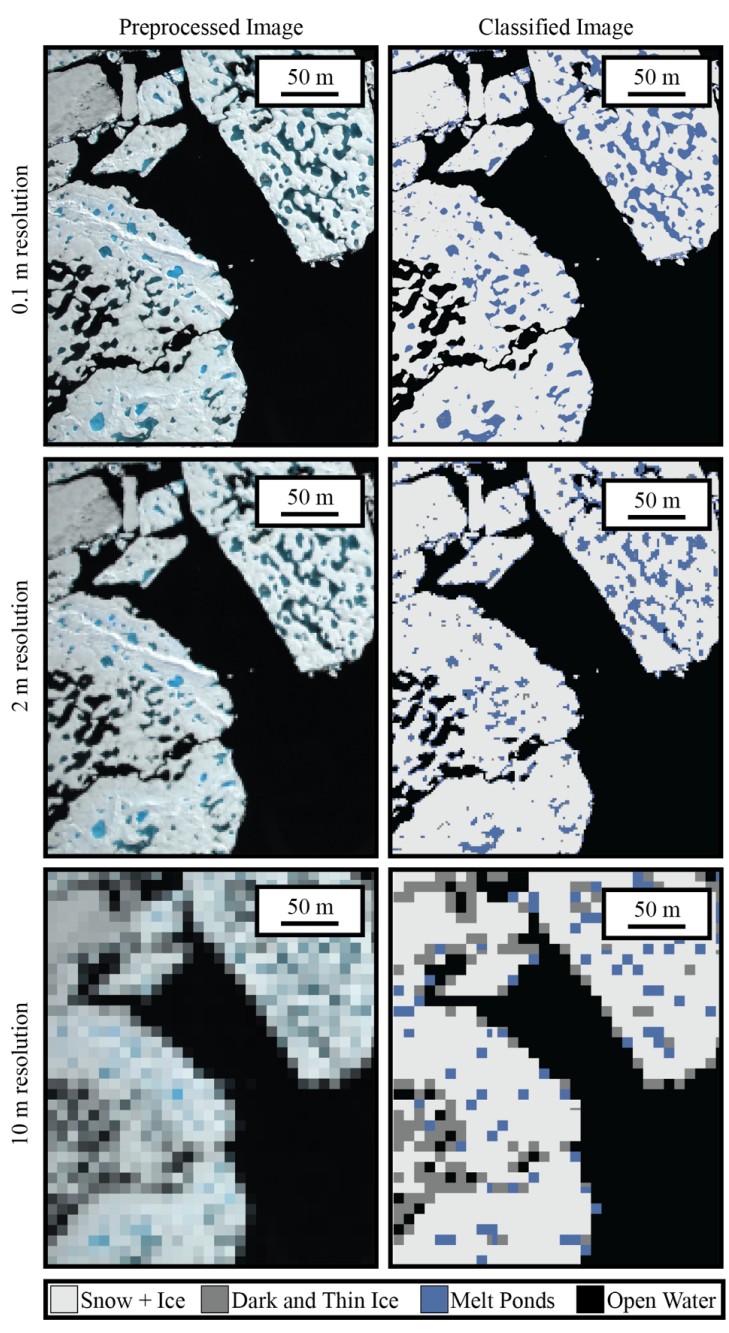


**Figure 12. Visual demonstration of the downsampling effect on a NASA IceBridge image. The top image is shown at the original 0.1 m resolution. The middle image is the equivalent resolution of a multispectral WorldView image without pansharpening. In the bottom image pixel size has begun to exceed the average melt pond size.**







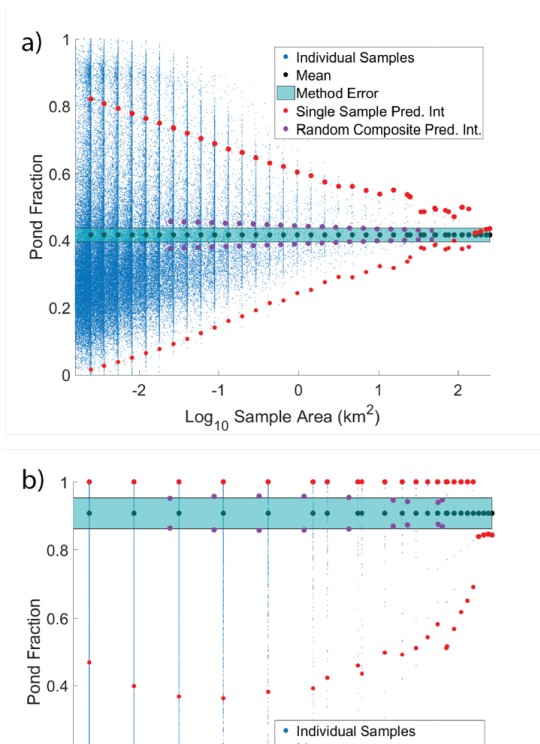


**Figure 13. Convergence of melt pond fraction (a) and ice fraction (b) for a WorldView image collected 25 June 2014 at 72°N**
**128°W as the area evaluated is increased. Small blue dots represent individual image subsets. For segments of a given size,**
**black dots represent the mean value of those samples, red dots represent the 95% prediction interval, and purple dots show**
**the 95% prediction interval for the same total area, but calculated from 100 randomly placed, smaller, samples. Cyan shaded**
**area represents the error in determination expected from the processing method.**




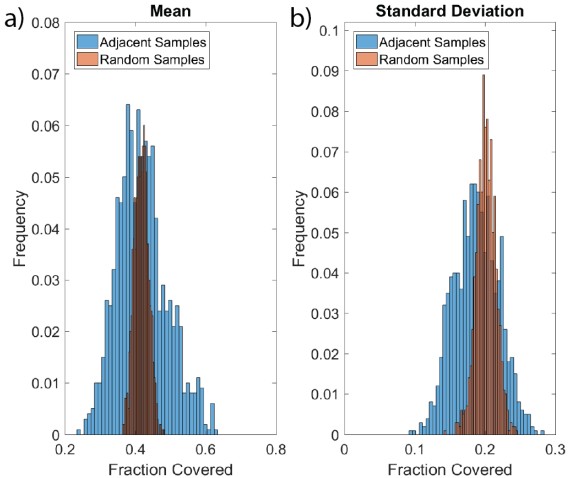

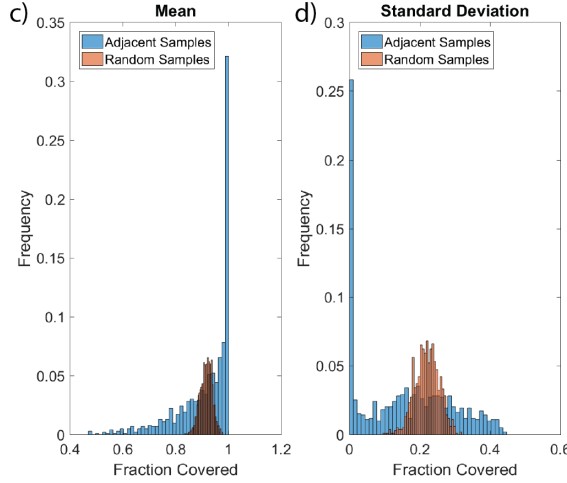

**Figure 14.** Histogram of mean (a) and standard deviation (b) of 1000 melt pond fraction estimates, each calculated from 100 sample areas on a 25 June 2014 WorldView image. The 100 samples were either randomly distributed across the image (red) or adjacent to each other (blue). Panels (c) and (d) show the same as (a) and (b), respectively, for ice fraction rather than melt pond fraction.





**Tables**

| Attribute | MS | PAN | Aerial |
|---|---|---|---|
| Mean (Pan) | light gray | blue | light gray |
| Mean (Coastal) | blue | light gray | light gray |
| Mean (Blue) | blue | light gray | blue |
| Mean (Green) | blue | light gray | blue |
| Mean (Yellow) | blue | light gray | light gray |
| Mean (Red) | blue | light gray | blue |
| Mean (Red Edge) | blue | light gray | light gray |
| Mean (NIR1) | blue | light gray | light gray |
| Mean (NIR2) | dark gray | light gray | light gray |
| Median (Pan) | light gray | blue | light gray |
| StDev (Pan) | light gray | blue | light gray |
| Min Intensity (Pan) | dark gray | blue | dark gray |
| Max Intensity (Pan) | dark gray | blue | dark gray |
| StDev (Blue) | dark gray | light gray | blue |
| StDev (Green) | dark gray | light gray | blue |
| StDev (Red) | dark gray | light gray | blue |
| Entropy | dark gray | blue | blue |
| Segment Size | dark gray | blue | blue |
| Image Date | blue | blue | blue |
| Coastal / Green | blue | light gray | dark gray |
| Blue / NIR1 | blue | light gray | light gray |
| Green / NIR1 | blue | light gray | light gray |
| Yellow / Red Edge | blue | light gray | light gray |
| Yellow / NIR1 | blue | light gray | light gray |
| Yellow / NIR2 | blue | light gray | light gray |
| Red / NIR1 | blue | light gray | light gray |
| (B1 - NIR1)/(B2 + NIR1) | blue | light gray | light gray |
| (G - R)/(G + R) | blue | light gray | light gray |
| (B - R)/(B + R)[1] | dark gray | light gray | blue |
| (B - G)/(B + G)[1] | dark gray | light gray | blue |
| (G - R)/(2*B - G - R)[1] | dark gray | light gray | blue |
| Neighbor Mean | blue | blue | blue |
| Neighbor StDev | dark gray | blue | blue |
| Neighbor Max | dark gray | blue | blue |
| Neighbor Entropy | dark gray | blue | blue |

[1]**Miao et al. 2015**
**Table 1. Attributes used for classifying each of the three image types. Blue squares indicate attributes that were used for**
**that image, dark gray squares indicate attributes that are available, but were not found to be sufficiently beneficial in the**
**classification to merit inclusion under our criteria. Light gray squares are ones where the attribute is not available on that**
**image type (e.g. band ratios on a panchromatic image). NIR are the near infrared wavelengths. B1 is the costal WorldView**
**band, and B2 is the blue band. R, B, and G, stand for red, green, and blue, respectively.**




| Image ID | Sensor Type | Date Collected | I+S | DTI | MP | OW | Accuracy |
|---|---|---|---|---|---|---|---|
| 102001002C214D00 | Panchromatic | 11-Mar-14 | 96 | 3 | 0 | 2 | 97 |
| 103001002E8F0D00 | Panchromatic | 18-Mar-14 | 97 | 3 | 0 | 0 | 97 |
| 102001002BBA0C00 | Panchromatic | 19-Mar-14 | 97 | 2 | 0 | 1 | 96 |
| 103001002FC75200 | Panchromatic | 23-Mar-14 | 94 | 4 | 0 | 3 | 95 |
| 102001002CB77C00 | Panchromatic | 27-Mar-14 | 98 | 2 | 0 | 0 | 100 |
| 1030010030403A00 | Panchromatic | 31-Mar-14 | 95 | 2 | 0 | 3 | 98 |
| 1030010031B65000 | Panchromatic | 4-Apr-14 | 96 | 3 | 0 | 1 | 99 |
| 102001002BA6C100 | Panchromatic | 8-Apr-14 | 93 | 3 | 0 | 4 | 100 |
| 103001002F79A700 | Panchromatic | 21-Apr-14 | 93 | 3 | 0 | 4 | 98 |
| 103001003037 1B00 | Panchromatic | 24-Apr-14 | 93 | 7 | 0 | 0 | 98 |
| 103001003102A600 | Panchromatic | 4-May-14 | 76 | 16 | 0 | 8 | 98 |
| 102001003007FA00 | Panchromatic | 13-May-14 | 87 | 3 | 0 | 10 | 97 |
| 10300100306F2E00 | Panchromatic | 19-May-14 | 83 | 4 | 0 | 13 | 96 |
| 102001003035D700 | Panchromatic | 13-Jun-14 | 49 | 7 | 25 | 18 | 95 |
| 1030010033AAC400 | Panchromatic | 19-Jun-14 | 20 | 3 | 16 | 61 | 97 |
| 1020010031DF9E00 | Panchromatic | 20-Jun-14 | 27 | 2 | 31 | 39 | 96 |
| 1020010032B94E00 | Panchromatic | 24-Jun-14 | 45 | 2 | 41 | 11 | 95 |
| 102001003122A700 | Panchromatic | 25-Jun-14 | 48 | 1 | 37 | 13 | 97 |
| 102001002F4F1A00 | Panchromatic | 28-Jun-14 | 57 | 2 | 28 | 14 | 95 |
| 10300100346D1200 | Panchromatic | 1-Jul-14 | 38 | 0 | 23 | 39 | 97 |
| 1030010035C8D000 | Panchromatic | 12-Jul-14 | 0 | 0 | 0 | 100 | 100 |
| 103001003421AB00 | Panchromatic | 20-Aug-14 | 0 | 0 | 0 | 100 | 100 |
| 10300100324B7D00 | Multispectral | 13-Jun-14 | 44 | 7 | 29 | 19 | 96 |
| 1030010033AAC400 | Multispectral | 19-Jun-14 | 16 | 3 | 19 | 62 | 97 |
| 10300100346D1200 | Multispectral | 1-Jul-14 | 44 | 2 | 26 | 28 | 98 |
| 1030010035C8D000 | Multispectral | 12-Jul-14 | 0 | 0 | 0 | 100 | 100 |
| 2016_07_13_05863 | IceBridge | 13-Jul-16 | 50 | 2 | 34 | 14 | 92 |
| 2016_07_13_05882 | IceBridge | 13-Jul-16 | 72 | 1 | 26 | 0 | 97 |
| 2016_07_13_05996 | IceBridge | 13-Jul-16 | 70 | 2 | 28 | 0 | 95 |
| 2016_07_13_06018 | IceBridge | 13-Jul-16 | 61 | 2 | 36 | 1 | 91 |
| 2016_07_13_06087 | IceBridge | 13-Jul-16 | 66 | 1 | 33 | 0 | 99 |
| 2016_07_16_00373 | IceBridge | 16-Jul-16 | 9 | 0 | 2 | 89 | 100 |
| 2016_07_16_00385 | IceBridge | 16-Jul-16 | 66 | 1 | 14 | 20 | 98 |





| | | | | | | | |
|---|---|---|---|---|---|---|---|
| 2016_07_16_00662 | IceBridge | 16-Jul-16 | 49 | 1 | 16 | 35 | 98 |
| 2016_07_16_00739 | IceBridge | 16-Jul-16 | 67 | 2 | 25 | 6 | 97 |
| 2016_07_16_01569 | IceBridge | 16-Jul-16 | 22 | 0 | 7 | 71 | 97 |
| 2016_07_16_02654 | IceBridge | 16-Jul-16 | 35 | 0 | 10 | 54 | 95 |
| 2016_07_19_01172 | IceBridge | 19-Jul-16 | 62 | 0 | 14 | 24 | 90 |
| 2016_07_19_01179 | IceBridge | 19-Jul-16 | 57 | 0 | 10 | 32 | 95 |
| 2016_07_19_02599 | IceBridge | 19-Jul-16 | 51 | 0 | 7 | 43 | 99 |
| 2016_07_19_02603 | IceBridge | 19-Jul-16 | 69 | 0 | 9 | 22 | 99 |
| 2016_07_19_02735 | IceBridge | 19-Jul-16 | 74 | 0 | 25 | 0 | 100 |
| 2016_07_19_03299 | IceBridge | 19-Jul-16 | 57 | 0 | 8 | 35 | 96 |
| 2016_07_21_01221 | IceBridge | 21-Jul-16 | 49 | 0 | 4 | 47 | 97 |
| 2016_07_21_01311 | IceBridge | 21-Jul-16 | 87 | 1 | 5 | 7 | 95 |
| 2016_07_21_01316 | IceBridge | 21-Jul-16 | 92 | 0 | 4 | 4 | 99 |
| DSC_0154 | Aerial sRGB | 8-Jun-09 | 43 | 4 | 53 | 0 | 94 |
| DSC_0327 | Aerial sRGB | 8-Jun-09 | 33 | 3 | 63 | 0 | 90 |
| DSC_0375 | Aerial sRGB | 8-Jun-09 | 96 | 0 | 4 | 0 | 99 |
| DSC_0422 | Aerial sRGB | 8-Jun-09 | 88 | 0 | 11 | 0 | 98 |
| DSC_0223 | Aerial sRGB | 10-Jun-09 | 46 | 1 | 53 | 0 | 93 |
| DSC_0243 | Aerial sRGB | 10-Jun-09 | 59 | 1 | 40 | 1 | 98 |
| DSC_0314 | Aerial sRGB | 10-Jun-09 | 89 | 0 | 11 | 0 | 95 |
| DSC_0319 | Aerial sRGB | 10-Jun-09 | 75 | 2 | 19 | 4 | 88 |
| DSC_0323 | Aerial sRGB | 10-Jun-09 | 37 | 2 | 61 | 0 | 95 |
| DSC_0338 | Aerial sRGB | 10-Jun-09 | 83 | 2 | 15 | 1 | 95 |
| DSC_0386 | Aerial sRGB | 10-Jun-09 | 80 | 3 | 14 | 3 | 89 |
| DSC_0394 | Aerial sRGB | 10-Jun-09 | 79 | 2 | 10 | 9 | 95 |
| DSC_0412 | Aerial sRGB | 10-Jun-09 | 63 | 2 | 24 | 10 | 92 |
| DSC_0425 | Aerial sRGB | 10-Jun-09 | 56 | 2 | 17 | 24 | 97 |
| DSC_0439 | Aerial sRGB | 10-Jun-09 | 71 | 1 | 6 | 22 | 98 |
| DSC_0441 | Aerial sRGB | 10-Jun-09 | 57 | 0 | 4 | 38 | 98 |
| DSC_0486 | Aerial sRGB | 10-Jun-09 | 53 | 1 | 17 | 29 | 96 |
| DSC_0634 | Aerial sRGB | 10-Jun-09 | 72 | 1 | 14 | 12 | 96 |
| DSC_0207 | Aerial sRGB | 13-Jun-09 | 80 | 1 | 19 | 0 | 96 |
| DSC_0514 | Aerial sRGB | 13-Jun-09 | 86 | 1 | 13 | 0 | 97 |

**Results Table 2. The complete results of imagery processed for this analysis. Descriptions for each image includes the image**
**type, date collected, the percent of the image that falls into each of the four categories, and the accuracy assessment.**




| Image Source | Training Dataset Size | Out-of-bag Error |
|---|---|---|
| Panchromatic WorldView | 1000 | 0.94 |
| Pansharpened WorldView | 859 | 0.89 |
| Aerial Imagery | 945 | 0.94 |
| IceBridge Imagery | 940 | 0.91 |

**Table 3. Out-of-Bag scores for the three training datasets used to classify imagery from each of the four sensor platforms.**