# Peer review of "Open Source Algorithm for Detecting Sea Ice Surface Features in"

_The Cryosphere, 2017_

## Referee Comment (RC1) · Anonymous Referee #1 · 21 Oct 2017

The paper describes and evaluates a tool for classification of high resolution satellite- and air-borne remotely sensed imagery. The tool includes open source software and a library of training data. The idea is that this tool is a community resource that can be built upon. This is a commendable effort. Such tools are badly needed. While the segmentation and classification algorithms have been applied to sea ice classification before by Miao et al (2015), as the authors acknowledge, the development of an end-to-end software package, and perhaps more importantly, the creation of training data sets are a valuable contribution. Furthermore, the authors attempt to address the problem of sample representativeness as it pertains to deriving metrics for analysis of sea ice surface characteristics. As the authors state, this is a start of the discussion, hopefully

community resources as, developed here, can help to move this discussion along.

I heartily recommend publishing the paper in The Cryosphere. However, I do have a few issues that the authors should address in a revised version of the manuscript.

General ——— The paper is well written. The figures, in particular the captions need some work. My opinion is that figures and captions should be stand alone, such that a reader should be able to understand what each figure is without reading the text. In some cases adding a key or a description of symbols in the figure caption would achieve this goal. For example, Figure 3 has no key for the classified image.

The section that needs the most attention is

Specific ——— Line 50. Missing a reference to Fetterer and Unterstiener (1998).

Line 63. When discussing alternative classification methods, it would be good to enumerate those methods applied to classification of sea ice. Furthermore, the authors give maximum likelihood classification as an example or an unsupervised algorithm. Maximum likelihood can be a supervised algorithm.

Section 4.1 ————- In the analysis of seasonal evolution of surface characteristics, my guess is that the field of view does not contain the same ice. The authors should make this clear or state why they think it is the same floe/ice. How fast is the ice likely to be moving at this location?

Line 433. I disagree with the statement that misclassification means that the algorithm fails to replicate human decision making. That might be the goal but one that is impossible to reach. To my mind, misclassification indicates that the algorithm doesn't give the same answer as a human would.

Line 531. This analysis is interesting but does it apply to the image used or to all images in general. How might the result change through time or with season?

Line 558. The Central limit Theorem is a mathematical theorem complete with proof.

I wouldn't say that it can be tested. What you are doing here is evaluating if you can predict the regional/image mean from a set of smaller samples/local means. One framework to evaluate this is hypothesis testing in which you pose the hypothesis that N sample means can predict/estimate the regional mean. This test applies the Central Limit Theorem but does not test it. This section needs to be reworked.

Line 559. The standard definition of the Central Limit Theorem is that independent variables can be added and normalized by $(X - mu)/(sigma/sqrt(n))$ to yield a normal distribution $N(0,1)$. Where X is the sample mean, mu population mean and sigma population standard deviation.

Figures ——— Check figure numbering

Figure 13. The cyan rectangle over the black dots make the dots look green (at least on my screen). The labels in the key need to match the description in the text. If some of the information is not discussed, I suggest removing it from the figure.

---

## Referee Comment (RC2) · Anonymous Referee #2 · 21 Oct 2017

The manuscript presents an algorithm for segmenting high-resolution hyperspectral, RGB, and panchromatic imagery into three classes: snow and ice, melt ponds and submerged ice, and open water. The algorithm appears to have good performance throughout the seasons, despite the challenging conditions during summer. Unique to this work is that the algorithm and training sets have been designed as an open source tool to invite and facilitate community use. This, together with the thorough error analysis by the authors, sets the stage for a successful, sustained effort from the community to extract much-needed sea ice information from a variety of high-resolution imagery. Overall, I find the manuscript to be well suited for the Cryosphere, and highly recommend publication once the following issues are addressed:

[Figure]

Line 50. I suggest adding Fetterer and Untersteiner (1998) and Arntsen et al. (2015) to the reference list.

Line 66. This isn't quite right. Several previous works have demonstrated surface feature retrievals from high-resolution imagery throughout the seasonal evolution of ice surface conditions.

Lines 136+. How does the algorithm differ from that in Miao et al? Please describe any differences.

Line 236. Does this melt pond definition include or exclude melt ponds that are melted through? Relative to previous works, is it typical to include submerged ice in the melt pond class or is it unique to this approach?

Line 241. Submerged ice isn't a type of melt pond. Please clarify this point. It would be helpful to comment on the effects of submerged ice on melt pond statistics of area and geometry, especially for scenes of advanced melt in the marginal ice zone.

Line 245. Would this criterion also include sea ice darkened by sediment and algae during the melt season?

Lines 252 and 307. Please provide more details on the shadow detection step for panchromatic and multispectral imagery. Does it differ from previous works?

Line 290. Please describe how image dates are used in the classification scheme.

Lines 305/491. How does this step distinguish a neighboring ridge from snow-covered ice? It's not clear, does the algorithm identify ridges as a separate class?

Line 310. Here and elsewhere, trade-offs in computational expenses are mentioned. It would be helpful to give a ballpark estimate of the computational expense if possible, e.g., is it O(N) or O(Nˆ2)?

Line 313. It's surprising that the Literal Image Derived Products from the Global Fiducials Library have been excluded from this analysis, as these publicly available images

have been the data source for several analyses of high-resolution sea ice imagery (Arntsen, Fetterer, Kwok, Webster). Do the authors anticipate that users will find the algorithm suitable for processing this imagery given its radiometric inconsistencies? Why or why not?

Line 319. Are the different results between experienced and inexperienced users a matter of definition? For example, how do experienced and inexperienced users classify submerged ice near floe edges?

Lines 361+. It would be helpful to explicitly include submerged ice in the melt pond class throughout the text and figures. For example, instead of "Melt pond," please state "Melt pond and submerged ice" or "Melt pond + submerged ice."

Lines 406+. Is this an Eulerian or Lagrangian site? How do the authors distinguish changes due to spatial heterogeneity from seasonal melt progression?

Lines 508+. For the aggregate-scale analysis, what type of ice was present in the analyzed scenes? How might the results change based on the presence of different sea ice types?

Lines 760/Figure 4 & 779/Figure 8. I suggest presenting the pixel counts as percentages of the total pixels evaluated and providing the total pixel count in the caption for ease of reading.

Line 782/Figure 9. In the caption or text, please provide the average scene size.

Line 786/Figure 10. Please state whether this site was Eulerian or Lagrangian in the caption.

Line 791/Figure 11. I suggest adding the resolution size as a secondary x-axis on the top of the plots for ease of reading.

Technical comments:

Line 52. Morphology seems like the wrong word here.

Typos in Lines 170, 293, 324, 396, and 170.

Line 191. Boundary seems like a better word than interface since the ice-ocean interface can be interpreted as the ice floe bottom.

Line 283. Here or earlier, I suggest defining texture since it has multiple meanings.

Lines 798-802. Inconsistent font.
* * *

---

## Referee Comment (RC3) · Anonymous Referee #3 · 4 Nov 2017

The manuscript entitled "Open Source Algorithm for Detecting Sea Ice Surface Features in High Resolution Optical Imagery" offers an open source software package to process high resolution optical imagery of sea ice, so that providing a standardized, automated, and reproducible technique and sea ice products. This work is valuable and significant in polar research community. I especially appreciate authors provide a complete set of training sample on Github, and even an interface to collect more user-input training samples. This reflects the true spirit of Open Source. I would like to recommend this manuscript to be published after author address my several major concerns as follows – 1) Please note that there is a previous effort to realize an open source package for sea ice feature detection: Sea Ice Imagery Classification with Machine

Learning and High-Performance Computing, XSEDE 2016 Polar Compute Hackathon - Sea Ice Team, Contributers: Alek Petty, Andrew Barrett, Xin Miao, Phil McDowall, Vivek Balasubramanian, https://github.com/polar-computing/SeaIce

Is there any relationship between the author's package with the XSEDE 2016 package? Please cite it if necessary.

2) L9: What is "dm-scale"? 3) The terms used in manuscript are not consistent. L13: . . .melt ponds and submerged ice, so this is definition used in (Miao, 2016). This includes two subclasses: melt pond (MP) and coastal submerged ice. So how did author calculate MP coverge in L364? How to extract MP from the broad category of "melt ponds and submerged ice"? Please make it clear.

4) L165: Did author consider the possible image distortion due to tilting of sRGB and other images?

5) L191&L255: Did author consider the shadow issues? Shadow is an interesting sea ice feature, please refer to Xin Miao, Hongjie Xie, Stephen F. Ackley, Songfeng Zheng, "Object-Based Arctic Sea Ice Ridge Detection From High-Spatial-Resolution Imagery", IEEE Geoscience and Remote Sensing Letters, 13(6): 787-791, 2016.

6) L225: One of the major feature of RF is that it only need a small amount of samples, very suitable in labor-intensive remote sensing project like sea ice detection.

7) L236: How did you separate regular melt pond (fresh water) and melt-through MP (salt water)? Why not use the technique provided in (Miao, 2016)?

8) L 256: I think it makes sense to combine 3.3.4 and 3.3.5 to 3.3.3.

9) L307: Can you provide an example? I don't understand why.

10) L367: Section 3.6 is very confusing. What do you mean by "larger sample"? Is it "larger number of samples"? What is "metric" here? L374: you mean 'observer' not 'user', rite?

11) L381: Figure 7 refer to Figure 8? Very confusing here.

12) L 389: Fig. 8??

13) L405: Only 4.1 not 4.2? Then author could remove this subtitle.

14) Comment: L596: Very positive contribution by sharing the training set!
* * *

---

## Author Comment (AC1) · 18 Dec 2017

[revised manuscript text omitted]

[1]**Miao et al. 2015**

[revised manuscript text omitted]

We next test the central limit theorem to see how well we can predict the error bounds from processing a single set of independent (i.e. randomly distributed) samples. The central limit theorem states that when taking the mean of a sufficiently large number of independent samples of a random variable, the standard error of the mean of the samples is equal to $\frac{\sigma}{\sqrt{N}}$ where σ is the standard deviation of the sample values and $N$ is the sample size. The standard deviation of pond coverage fraction in sets of 100 sub-images ranged from 0.15 to 0.25 across the 1000 sample sets run (see histogram in Fig. 14b) This yields a predicted standard error of the mean determined from any one of these sets of 0.015 to 0.025. The observed standard deviation in the mean values across all 1000 sample sets presented in Fig. 14a is 0.0201, indicating that the central limit theorem applies in this case.

Returning

| **Page 17: [2] Deleted** | **Nicholas C. Wright** | **12/18/17 12:39:00 PM** |

, this time representing twice the standard error determined from the central limit theorem.

| **Page 17: [3] Deleted** | **Nicholas C. Wright** | **12/18/17 12:39:00 PM** |

permits expedient determination of melt pond fraction within that image area with small error bounds. If the total image is large enough, the value will be representative of the

| **Page 17: [4] Deleted** | **Nicholas C. Wright** | **12/18/17 12:39:00 PM** |

scale. In this case, processing as little as 5km$^2$ (~0.5%) of the image permits determination of a mean that lies within 0.025 of the true image mean 95% of the time. Also indicated on the plot is a 5% uncertainty band around the mean

| **Page 17: [5] Deleted** | **Nicholas C. Wright** | **12/18/17 12:39:00 PM** |

A test of the central value theorem again shows that it also applies in this case and provides a good estimate of the error of a mean ice fraction calculated from a set of random sub images. The green dots again indicate

| **Page 17: [6] Deleted** | **Nicholas C. Wright** | **12/18/17 12:39:00 PM** |

must process imagery representing at least 5km$^2$ in surface area, selected in at least

| **Page 25: [7] Deleted** | **Nicholas C. Wright** | **12/18/17 12:39:00 PM** |

**Important**

| **Page 25: [7] Deleted** | **Nicholas C. Wright** | **12/18/17 12:39:00 PM** |

**Important**

| **Page 25: [7] Deleted** | **Nicholas C. Wright** | **12/18/17 12:39:00 PM** |

**Important**

| **Page 25: [7] Deleted** | **Nicholas C. Wright** | **12/18/17 12:39:00 PM** |

**Important**

| **Page 25: [7] Deleted** | **Nicholas C. Wright** | **12/18/17 12:39:00 PM** |

**Important**

| Page 25: [7] Deleted | Nicholas C. Wright | 12/18/17 12:39:00 PM |

**Important**

| Page 25: [7] Deleted | Nicholas C. Wright | 12/18/17 12:39:00 PM |

**Important**

| Page 25: [7] Deleted | Nicholas C. Wright | 12/18/17 12:39:00 PM |

**Important**

| Page 25: [7] Deleted | Nicholas C. Wright | 12/18/17 12:39:00 PM |

**Important**

| Page 25: [7] Deleted | Nicholas C. Wright | 12/18/17 12:39:00 PM |

**Important**

| Page 25: [7] Deleted | Nicholas C. Wright | 12/18/17 12:39:00 PM |

**Important**

| Page 25: [8] Deleted | Nicholas C. Wright | 12/18/17 12:39:00 PM |

**patterns**

| Page 25: [8] Deleted | Nicholas C. Wright | 12/18/17 12:39:00 PM |

**patterns**

| Page 25: [8] Deleted | Nicholas C. Wright | 12/18/17 12:39:00 PM |

**patterns**

| Page 25: [8] Deleted | Nicholas C. Wright | 12/18/17 12:39:00 PM |

**patterns**

| Page 25: [8] Deleted | Nicholas C. Wright | 12/18/17 12:39:00 PM |

**patterns**

---

## Author Comment (AC2) · 18 Dec 2017

Reply to Anonymous Referee #1 from 21 Oct 2017

*Note: Author responses are in plain text following the original referee comment shown in italicized text.*

*The figures, in particular the captions need some work. My opinion is that figures and captions should be stand alone, such that a reader should be able to understand what each figure is without reading the text. In some cases adding a key or a description of symbols in the figure caption would achieve this goal. For example, Figure 3 has no key for the classified image.*

> We have added a classification key to Figure 3. Thank you for the suggestion – we have also incorporated edits throughout to make image captions more complete and descriptive.

*Line 50. Missing a reference to Fetterer and Untersteiner (1998).*

> We have included this reference as well as one to Arntsen et al. 2015.

*Line 63. When discussing alternative classification methods, it would be good to enumerate those methods applied to classification of sea ice. Furthermore, the authors give maximum likelihood classification as an example or an unsupervised algorithm. Maximum likelihood can be a supervised algorithm.*

> Each reference on line 63 details a classification method applied to sea ice. We have edited line 98 to indicate that those references refer to a method applied to image processing generally, and not specifically to the classification of sea ice.

> It is true that maximum likelihood classifiers can be supervised in some cases - we have revised this discussion of unsupervised classification algorithms to be more precise.

*Section 4.1. In the analysis of seasonal evolution of surface characteristics, my guess is that the field of view does not contain the same ice. The authors should make this clear or state why they think it is the same floe/ice. How fast is the ice likely to be moving at this location?*

> You are correct – the ice seen at this location is not a single floe. This explains the sudden increase and then subsequent decrease in open water fraction in late June, as well as the completely ice-free water by August. We have included the following sentence in Section 4.2 to clarify that we are looking at Eulerian rather than Lagrangian view: "The site is Eulerian; it observes a single location in space rather than follow a single ice floe through its lifecycle as it drifts".

*Line 433. I disagree with the statement that misclassification means that the algorithm fails to replicate human decision making. That might be the goal but one that is impossible to reach. To my mind, misclassification indicates that the algorithm doesn't give the same answer as a human would.*

> We agree that we are not replicating the decision-making process, but rather the end result. We have revised this section to clarify our definition of misclassification. Line 474 was changed to: "The algorithm either assigned the same classification as a human would have, or it did not", and line 478 has been rewritten to: "The first type of internal error is misclassification error, where the image classification algorithm fails to assign the same classification that a human expert would choose".

*Line 531. This analysis is interesting but does it apply to the image used or to all images in general. How might the result change through time or with season?*

> This analysis applies specifically to the image used, though there is nothing particularly unique about the image analyzed. We believe the results are applicable to all images in general, but a complete demonstration of that is better suited to its own study, and we suggest future work to expand this analysis to a general rule. The statistical methods that we use here should be independent of the seasonality of ice (so long as the metric you are investigating can by captured by the image scale, e.g. this works for melt pond fraction and not for ice fraction). At the core, we are just using a sample of some size as a means to estimate a population statistic.

> We have added line 598 to detail this in the manuscript: "The statistical approach for determining image statistics should not depend on the seasonality of the image nor the type of image used so long as the total area observed is sufficiently greater than the variability in the surface feature being investigated."

*Line 558. The Central limit Theorem is a mathematical theorem complete with proof. I wouldn't say that it can be tested. What you are doing here is evaluating if you can predict the regional/image mean from a set of smaller*

*samples/local means. One framework to evaluate this is hypothesis testing in which you pose the hypothesis that N sample means can predict/estimate the regional mean. This test applies the Central Limit Theorem but does not test it. This section needs to be reworked.*

*Line 559. The standard definition of the Central Limit Theorem is that independent variables can be added and normalized by (X - mu)/(sigma/sqrt(n)) to yield a normal distribution N(0,1). Where X is the sample mean, mu population mean and sigma population standard deviation.*

These are good points. We have significantly revised this section. The takeaway message remains largely the same as in the original version, but both the text and methodology has been improved for new version. Instead of applying the Central Limit Theorem directly, we instead analyze the sample size required to estimate the regional mean, and address the difference in measuring from spatially correlated samples versus randomly selected samples.

*Figures: Check figure numbering*

Thanks for catching this – we have fixed the figure numbering issue.

*Figure 13. The cyan rectangle over the black dots make the dots look green (at least on my screen). The labels in the key need to match the description in the text. If some of the information is not discussed, I suggest removing it from the figure.*

We have fixed the coloration issue with the mean dots.

---

## Author Comment (AC3) · 18 Dec 2017

Reply to Anonymous Referee #2 from 21 Oct 2017

*Note: Author responses are in plain text following the original referee comment shown in italicized text.*

*Line 50. I suggest adding Fetterer and Untersteiner (1998) and Arntsen et al. (2015) to the reference list.*

We have added both references to this section. Although the Arntsen et al. 2015 paper uses the method presented in Miao et al. 2015.

*Line 66. This isn't quite right. Several previous works have demonstrated surface feature retrievals from high-resolution imagery throughout the seasonal evolution of ice surface conditions.*

We have reviewed a large number of previous works detailing the classification of high resolution optical imagery of sea ice (e.g. Arntsen et al., 2015; Fetterer and Untersteiner, 1998; Inoue et al., 2008; Kwok, 2014; Lu et al., 2010; Miao et al., 2015; Perovich et al., 2002b; Renner et al., 2014; Webster et al., 2015), and have not come across any that are demonstrated on a complete seasonal melt cycle. To the best of our knowledge no such work exists. We will gladly incorporate further information into this section if the reviewer can point us to the references they are referring to.

*Lines 136+. How does the algorithm differ from that in Miao et al? Please describe any differences.*

While this algorithm is inspired by the work of Miao et al. 2015 to use image segmentation followed by classification with a random forest algorithm, the implementation of that workflow is quite different. To convey this, we have added line 132: "Our implementation of the segmentation and classification, however, were custom-built using well known image processing tools (Pedregosa et al., 2011, van der Walt et al., 2014) in an open source format".

The algorithm presented by Miao et al. (2015) uses the ENVI GIS software package. As such, there are some specifics that remain proprietary to ENVI. Where we know how the Miao et al. algorithm behaves, we have stated the similarities and differences. We use a custom-built segmentation technique (section 3.2) that is different than the Miao et al. method. In the random forest machine learning technique, we use some attributes that were developed by Miao et al. (2015) (attributed in line 267 and 303), as well as attributes new to our method (lines 302-313).

*Line 236. Does this melt pond definition include or exclude melt ponds that are melted through? Relative to previous works, is it typical to include submerged ice in the melt pond class or is it unique to this approach?*

Our melt pond definition, which is provided in lines 239-248. excludes the area of melt ponds that has melted through completely (see Figure 5). Our approach to the surface classifications was to consider primarily shortwave optical properties. Submerged ice and melt ponds have similar optical properties and impact the solar energy balance in the same way. Thus it makes sense to group them into a single category. Previous works have taken both approaches. Miao et al. (2015), for example, presented a method for distinguishing general submerged ice from contained melt ponds by analyzing their proximity to open water. However, separating these classes is not necessary for all applications. Spectral unmixing algorithms, such as those presented by Rosel et al. 2012, to determine melt pond fraction on a larger scale consider only aggregate optical properties, and melt pond fraction would necessarily include the general submerged ice category as well.

*Line 241. Submerged ice isn't a type of melt pond. Please clarify this point. It would be helpful to comment on the effects of submerged ice on melt pond statistics of area and geometry, especially for scenes of advanced melt in the marginal ice zone.*

*Lines 361+. It would be helpful to explicitly include submerged ice in the melt pond class throughout the text and figures. For example, instead of "Melt pond," please state "Melt pond and submerged ice" or "Melt pond + submerged ice."*

To be more clear on our definition, we have changed this category to be "melt pond and submerged ice" throughout the figures and text.

Our hope is to spur community discussion with these surface type definitions, and so we have presented what we feel is the most widely applicable way to standardize 'ponded ice'. We acknowledge in line 236 that there are different opinions. We found that many experts in the sea ice community have subtly

different definitions of the surface types, even beyond the distinction being made here. (as we discovered when producing the data for Figure 4.) From a shortwave optical stand point, submerged ice and melt ponds are functionally the same, and since radiative balance is a primary reason to study ponds, we argue in line 244-245 that it makes sense to group them as a single category. For a study concerned with pond geometry this is obviously not the case, and there are methods (such as those discussed in Miao et al. 2015) to separate general submerged ice from melt ponds. These could easily be applied to our output by an interested user.

*Line 245. Would this criterion also include sea ice darkened by sediment and algae during the melt season?*

Yes, though we have not seen either of these features in the images we processed for this paper.

*Lines 252 and 307. Please provide more details on the shadow detection step for panchromatic and multispectral imagery. Does it differ from previous works?*

We did not use the shadow category for multispectral imagery. There is not a separate step for shadow detection, per se, rather an additional training category for the machine learning algorithm. We have edited the discussion of the shadow category in section 3.3.2 to reflect that we are not presenting shadows as a classification category. We have also added lines 258-261 to illustrate the differences to previous approaches to handling shadows.

This step does differ from previous work. In Webster et al. (2015), for example, ridge shadows are directly masked and set to the maximum pixel values. Our approach also differs from that in Miao et al. (2015), as our shadow class is not an independent classification in the output and it is only used for images prior to melt onset. Miao et al. (2016) details a more sophisticated ridge and shadow detection scheme

*Line 290. Please describe how image dates are used in the classification scheme.*

We have edited lines 296-300 to clarify how image dates are used for classification. The image acquisition dates are an attribute that the random forest can use to make a prediction. Image date is a simple means of estimating melt state, which improves the ability of the classifier to correctly predict surface conditions.

*Lines 305/491. How does this step distinguish a neighboring ridge from snow-covered ice? It's not clear, does the algorithm identify ridges as a separate class?*

Line 491 incorrectly implied that we are detecting ridges directly. We've edited line 491 to clear up this point. While we have methods for indirect detection of ridges (i.e. their shadows – see revised lines 258-261) we do not distinguish ridges from snow covered ice. We have reworked lines 309-318 to illustrate that bright ridge pixels are an example of the benefit of looking at object neighbors, and not a method for creating a ridge class.

*Line 310. Here and elsewhere, trade-offs in computational expenses are mentioned. It would be helpful to give a ballpark estimate of the computational expense if possible, e.g., is it O(N) or O(N^2)?*

The algorithm is roughly O(N), but it is difficult to quantify the computational expense in big-O terms for this application. High resolution satellite images are quite large, and can easily have millions of image objects. Therefore, any small increase in the time required to evaluate each image object (such as a more complex neighbor analysis) dramatically increases the total processing time.

We have edited lines and 197, 274 to refine our meaning behind 'computational expense'.

*Line 313. It's surprising that the Literal Image Derived Products from the Global Fiducials Library have been excluded from this analysis, as these publicly available images have been the data source for several analyses of high-resolution sea ice imagery (Arntsen, Fetterer, Kwok, Webster). Do the authors anticipate that users will find the algorithm suitable for processing this imagery given its radiometric inconsistencies? Why or why not?*

We do not anticipate any issues with the NTM imagery from the Global Fiducials Library, and 1m resolution is high enough to get good results (see figures 11 and 12). From an image processing standpoint, the NTM imagery is very similar to panchromatic WorldView imagery, and we therefore do not believe processing the NTM imagery would change the discussions of this paper. In unpublished work Arntsen has tested this algorithm on the NTM imagery with success.

We have added line 151: "The imagery sources chosen for this analysis were selected to be representative of the variation that exists in optical imagery of sea ice, but there is an abundance of image data that can be processed with this technique."

*Line 319. Are the different results between experienced and inexperienced users a matter of definition? For example, how do experienced and inexperienced users classify submerged ice near floe edges?*

The experience and inexperienced users had the same classification definitions in front of them as they worked their way through the training sets. Though some users might have had different opinions of the surface types on their own, the lack of standard definition is not the reason for disagreement. The definitions of the ice types, including for example submerged ice, were set in advance and provided to all users. The differences arise from the user's ability to interpret the definitions and apply them. We have added a sentence in the paper clarifying this point (line 329). As we established the definition of melt pond to include submerged ice on the edge of a floe ahead of time, users were consistent in their classification of these categories.

*Lines 406+. Is this an Eulerian or Lagrangian site? How do the authors distinguish changes due to spatial heterogeneity from seasonal melt progression?*

*Line 786/Figure 10. Please state whether this site was Eulerian or Lagrangian in the caption.*

The site is Eulerian. We have clarified this in the relevant figure captions and added line 505: "The site is Eulerian; it observes a single location in space and does not follow a single ice floe through its lifecycle as it drifts".

*Lines 508+. For the aggregate-scale analysis, what type of ice was present in the analyzed scenes? How might the results change based on the presence of different sea ice types?*

The images used in the aggregate scale analysis contained primarily first year ice in various stages of melt, and we have noted this in line 522. We have noted in the manuscript in lines 585/591 that this method applies only to melt pond fraction, as we discovered that the images were not large enough to accurately capture ice fraction. Within the melt pond fraction category, we do not believe a different ice type would substantially change the results (lines 594-596), as this analysis is at its core a statistical problem (how to estimate a population based on a sample).

*Lines 760/Figure 4 & 779/Figure 8. I suggest presenting the pixel counts as percentages of the total pixels evaluated and providing the total pixel count in the caption for ease of reading.*

That is a good suggestion. We have added that information to the figure.

*Line 782/Figure 9. In the caption or text, please provide the average scene size.*

Another good suggestion, which we have also added to the figure caption.

*Line 791/Figure 11. I suggest adding the resolution size as a secondary x-axis on the top of the plots for ease of reading.*

We have changed the x-axis to be in units of resolution in meters. The axis is still on a log scale, but as you suggested, it is much easier to read in this format.

*Line 52. Morphology seems like the wrong word here.*

Changed 'morphology of surface conditions' to 'morphology of surface features'. The morphology of a feature is its structure or form, and here we are discussing the difficulty of lower resolution optical sensors in directly observing the structure of surface features.

---

## Author Comment (AC4) · 18 Dec 2017

Reply to Anonymous Referee #3 from 04 Nov 2017

*Note: Author responses are in plain text following the original referee comment shown in italicized text.*

*Please note that there is a previous effort to realize an open source package for sea ice feature detection: Sea Ice Imagery Classification with Machine Learning and High-Performance Computing, XSEDE 2016 Polar Compute Hackathon - Sea Ice Team, Contributers: Alek Petty, Andrew Barrett, Xin Miao, Phil McDowall, Vivek Balasubramanian, https://github.com/polar-computing/SeaIce Is there any relationship between the author's package with the XSEDE 2016 package? Please cite it if necessary.*

> We had not seen that package prior to this review. Our work has no basis in the code referenced. We appreciate being informed about another activity and future collaboration between our efforts and this project may be beneficial to the community. Citation does not seem to be warranted since the code referenced does not appear to be published yet.

*L9: What is "dm-scale"?*

> 'dm' here stands for decimeter – the SI term for $10^{-1}$ meters.

*The terms used in manuscript are not consistent. L13: . . .melt ponds and submerged ice, so this is definition used in (Miao, 2016). This includes two subclasses: melt pond (MP) and coastal submerged ice. So how did author calculate MP coverge in L364? How to extract MP from the broad category of "melt ponds and submerged ice"? Please make it clear.*

> The definition we use for melt ponds is stated in section 3.3.2, line 243 to 244, and we are consistent with the usage throughout the manuscript. Our definition is not the same as that used in Miao et al. (2016), as we do not differentiate between melt ponds and costal submerged ice. We explicitly clarify this difference on lines 245-249. From a shortwave optical albedo standpoint it is unnecessary to separate these classes, and therefore we do not attempt to extract melt ponds according to this narrower definition. Other works present methods to separate these (Miao et al., 2015, for example) that could be applied to the results presented here for users interested in that application.

> We have changed the category name to be "melt ponds and submerged ice" to alleviate some of the confusion for this category.

*L165: Did author consider the possible image distortion due to tilting of sRGB and other images?*

> For this work, no. As we are not trying to answer any scientific questions based on this sRGB imagery specifically, we did not attempt to correct image distortions. The algorithm is able to classify images even with small amounts of off-nadir distortion. Applications that seek to use sRGB imagery to answer scientific questions should address any image distortion present. As sRGB imagery is not standard (unlike WorldView), correcting image distortion must be done on a case by case basis using positioning, pointing, and lens information which was not available in the data we worked with.

*L225: One of the major feature of RF is that it only need a small amount of samples, very suitable in labor-intensive remote sensing project like sea ice detection.*

> This is true: We have added "[…] even with relatively small training datasets" to line 228.

*L236: How did you separate regular melt pond (fresh water) and melt-through MP (salt water)? Why not use the technique provided in (Miao, 2016)?*

> We did not differentiate between fresh water melt ponds and salt water melt ponds. Our motivation lies in short wave optical properties of melt ponds and from that perspective the distinction between salt and fresh water is not important. We added line 241 to clarify this: "Our surface type definitions focus on the behavior of a surface in absorption of shortwave radiation and radiative energy transfer". However, melt ponds that are completely melted through were classified as open water based on their unique spectral characteristics (Figure 5).

*L 256: I think it makes sense to combine 3.3.4 and 3.3.5 to 3.3.3.*

> Thank you for the suggestion. We have combine sections 3.3.4 and 3.3.5 into a single section describing all of the attributes calculated for each image object.

*L191&L255: Did author consider the shadow issues? Shadow is an interesting sea ice feature, please refer to Xin Miao, Hongjie Xie, Stephen F. Ackley, Songfeng Zheng, "Object-Based Arctic Sea Ice Ridge Detection From High-Spatial-Resolution Imagery", IEEE Geoscience and Remote Sensing Letters, 13(6): 787-791, 2016.*

*L307: Can you provide an example? I don't understand why.*

In lines 259-265 we have edited the description of shadow detection to better illustrate our approach, and in lines 316-323 we have edited the description of detecting ridges in neighboring regions.

Shadows are an interesting feature of sea ice, but classification of shadow regions is beyond the scope of this paper. We are not trying to present shadow or ridge detection as a stand-alone feature. In spring panchromatic WorldView imagery, shadows look similar to melt ponds, and lines 259-265 is a simple method to address the similarity.

The example presented in line 316 (was 307) is an example of how using neighborhood statistics help identify the classification of an object, and we have reworded it this to be more apparent.

*L367: Section 3.6 is very confusing. What do you mean by "larger sample"? Is it "larger number of samples"? What is "metric" here? L374: you mean 'observer' not 'user', rite?*

We agree that this section as written was confusing, and we have restructured much of this paragraph to increase its clarity. The larger sample assessed the accuracy of 1000 pixels instead of 100. User and observer were referring to the same thing, and we have simplified this to a single term.

*L381: Figure 7 refer to Figure 8? Very confusing here.*

No, but there was a typo in describing Figure 7 here. This sentence has been edited to be more precise in its reference to Figure 7.

*L 389: Fig. 8??*

This line correctly references Figure 8. We have rearranged the order of this sentence to be more clear.

*L405: Only 4.1 not 4.2? Then author could remove this subtitle.*

We've added a section 4.1 here to divide this section into 4.1 and 4.2

*Comment: L596: Very positive contribution by sharing the training set!*

Thank you!

---

## Author Response (AR2)

1 Reply to Anonymous Referee #1 from 19 Jan 2018

2 *Note: Author responses are in plain text following the original referee comment shown in italicized text.*

4 ***Referee Comments:***
6 *With regards to estimate the required sample size using equation in line 568, I am not sure their calculation for*
7 *required number of samples is correct in the current revision - or at least that they have used incorrect values. This*
8 *needs to be addressed before publication as I think it alters the size of sample required for analysis.*
10 *I still think this paper should be published but the calculations need to be correct. Two things need to be done: 1) the*
11 *calculations and values used need to be checked to make sure they are correct. 2) They need to state what they mean*
12 *by margin of error (either % error of best estimate or absolute error) and the absolute error needs to be used in the*
13 *calculation of sample size.*
15 *My specific comments about the problems I see are here.*
17 *To find an acceptable sample size they use:*
19 *n = (z * sigma / margin_of_error)**2 (1)*
21 *where n is the number of samples required to get margin_of_error, z is the z-score (x_est - X/sigma), sigma is the*
22 *standard deviation and margin of error is, in this case, the accepted error of the measurement. They want to find n*
23 *such that the margin of error is 5% and appear to use a value of 0.05 in equation 1.*
25 *Based on the text you sent, it appears they use z=1.96, sigma=0.2 and margin_of_error=0.05. Using these values and*
26 *their equation (equation 1 here), I get n~62, not n=64 given in the paper. If I set z=2 (e.g. a 2*sigma margin of error),*
27 *I get n=64. They either need to revise n or change z to 2, in which case the confidence interval is 95.4%. This is not a*
28 *big deal (using z=1.96 versus z=2.0 does not change the sample size much), but it is frustrating if you try to follow*
29 *their calculations. The resulting sample size should be consistent with the parameters for the equation.*
31 *A bigger issue is how they use margin of error. They say they want to find a sample size that will give a 5% margin of*
32 *error with ~95% confidence interval. I take a margin of error of 5% to mean 5% of the best estimate of melt pond (or*
33 *ice) fraction. Based on Figure 14a, the best estimate of pond fraction for the image they use is ~0.4. The corresponding*
34 *margin of error (based on my understanding) is 0.02 (0.05*0.4). Using equation 1, this gives a sample size of almost*
35 *400, not the ~60 they give in the paper. A margin of error of 0.05 in terms of fraction is 12.5%. This means that more*
36 *than 6 times more samples are required to get the desired margin of error of 5%. I give my reasoning for this below.*
38 *Reasoning*
39 *---------------*
40 *Equation 1 follows from the statement of uncertainty of a measurement*
42 *x_est = x_best +/- margin_of_error (2)*
44 *where x_best is the mean of n measurements. margin_of_error can be given in the same units as x_best (here a*
45 *dimensionless fraction [0,1]) or a percentage. The percentage is 100*margin_of_error/x_best. margin_of_error is*
46 *given by*
48 *margin_of_error = t*standard_error = t*sigma/sqrt(n).*
50 *t is the number of standard errors a fraction P other measurements are expected to fall from x_best. In their case, P*
51 *is 95%, which corresponds to a t of 1.96. This is equivalent to a z-score of 1.96. margin_of_error, standard_error,*
52 *and sigma are all in the same units as x_best.*

**Author's Response:**
The reviewer is correct in noting our miscalculation of equation 1 (now labeled eqn. 3 in manuscript): With the
provided values, n should equal 62, not 64. Thank you for catching that error. Recalculating this equation led us to
realize a second error in the values used. Sigma, the standard deviation of the population, should be read from Fig.
14a, not Fig. 14b (Fig 14b is showing the standard deviation of a different parameter – not the population). Because
of this confusion we have chosen to remove Fig 14b/d from display in the hopes of simplifying our presentation. Based
on Figure 14a, sigma is closer to 0.05.
The reviewer's discussion of margin of error is also insightful. Here we meant a 'margin of error' of 5% as an absolute
error, not a relative one. That is, mpf = 0.40 +/- 0.05, not 0.40 +/- 5%. As the reviewer noted, this is the incorrect
usage of margin of error, and an absolute error of +/- 0.05 would correspond to a margin of error = 12.5% (0.05/0.4).
This paragraph has been reworked to incorporate the more appropriate margin of error = 0.016 (0.41 * 0.04). The new
margin of error and the correct standard deviation calculated from the 1000 samples measured to produce Fig. 14 are
used to revise the estimated number of samples required.

Reply to Anonymous Referee #2 from 07 Feb 2018

*Note: Author responses are in plain text following the original referee comment shown in italicized text.*

*The revised manuscript has greatly improved, and I thank the authors for giving careful attention to addressing the*
*raised concerns. The presented work makes a nice contribution to the field, and it has been a pleasure reviewing it. I*
*have remaining concerns that should be addressed before final publication:*

*Line 55. "but were historically limited" High resolution imagery is still spatially limited in basin-wide coverage. It's*
*incorrect to suggest that it's no longer an issue.*

We have added to line 58/59: "While high resolution imagery still does not provide basin-wide coverage,
[…]". However, new projects (e.g. Planet Labs) are already imaging the entire planet in high resolution
imagery (3m) on a daily timescale, so the spatial coverage may not be as much of an issue in the very near
future.

*Line 66. "none have been challenged by imagery collected across the seasonal evolution of the ice" Fetterer and*
*Untersteiner, 1998 studied the seasonal evolution of the ice surface from a Eulerian perspective, much like this*
*analysis. Rosel et al., 2012 studied the seasonal evolution of the ice surface from a basin-wide perspective. Perovich*
*et al., 2002, Arntsen et al., 2015, and Webster et al., 2015 studied the seasonal evolution of the ice surface from a*
*Lagrangian perspective. It's not clear why the authors are suggesting that this hasn't been done before with airborne*
*and satellite imagery.*

You are correct – this phrase now reads: "Furthermore, no single method has been used to process data from
multiple sensor platforms or documented and released for wide-spread community use."

*Lines 260-262. Is it correct that, once melt onset occurs, shadows are not accounted for? Please make this explicit,*
*as it will affect the interpretation of melt pond retrievals during early pond formation.*

No – shadows are accounted for in all imagery. However, they played the most significant role in pre-melt
panchromatic imagery. We have reworded lines 260-266 to clarify and note that any misclassifications of
shadows on ice interpreted as melt ponds (or other shadow related errors) would be accounted for in the
accuracy assessment.

*Lines 262-265. The approach in Webster et al. 2015 does not directly mask shadows as a separate class, but groups*
*them with the ice/snow category.*

Webster et al. 2015 does mask shadow regions from our understanding, but we have added an extra reference
to this paper to clarify that we are also grouping shadows with the ice/snow category.

*Lines 332-335. More details are needed here. The explanations should be more explicit, "We use image acquisition*
*dates as a proxy for melt onset." Which dates were used as thresholds for defining melt onset, and is there a latitudinal*
*dependency for these dates?*

We have added more details here describing how this attribute is used: It is a continuous variable in Julian
day format. We have also included your suggestion on line 305: "To ensure that the method remains fully
automated image acquisition date is used as a proxy for melt state, whereby larger Julian day values correlate
to later in the melt season." There is no threshold for defining melt onset, but rather a continuum from spring
to fall that the machine learning algorithm can consider in its classifications. As we note in the text there are
likely ways to improve this (and other) attributes in future work, but in the current work this attribute was
found to be beneficial in classifying segments.

[revised manuscript text omitted]